# A Closer Look at Smoothness in Domain Adversarial Training

## Abstract

Domain adversarial training has been ubiquitous for achieving invariant representations and is used widely for various domain adaptation tasks. In recent times, methods converging to smooth optima have shown improved generalization for supervised learning tasks like classification. In this work, we analyze the effect of smoothness enhancing formulations on domain adversarial training, the objective of which is a combination of task loss (eg. classification, regression etc.) and adversarial terms. In contrast to task loss, our analysis shows that *converging to smooth minima w.r.t. adversarial loss leads to sub-optimal generalization on the target domain*. Based on the analysis, we introduce the Smooth Domain Adversarial training (SDAT) procedure, which effectively enhances the performance of existing domain adversarial methods for both classification and object detection tasks. Our smoothness analysis also provides insight into the extensive usage of SGD over Adam in domain adversarial training.

## 1 Introduction

Domain Adversarial Training (Ganin & Lempitsky, 2015) (DAT) refers to adversarial learning of neural network based feature representations that are invariant to the domain. For example, car images from the clipart domain have similar feature representations as car images from the web domain. DAT has been widely useful in diverse areas (cited 3540 times) such as fairness (Adel et al., 2019), object detection (Saito et al., 2019), domain generalization (Li et al., 2018), image-to-image translation (Liu et al., 2017) etc. The prime driver of research on DAT is its application in unsupervised Domain Adaptation (DA), which aims to learn a classifier using labeled source data and unlabeled target data, such that it generalizes well on target data. Various enhancements like superior objectives (Acuna et al., 2021; Zhang et al., 2019), architectures (Long et al., 2018) etc. have been proposed to improve its effectiveness. However, as DAT objective is combination of Generative Adversarial Network (GAN) (Goodfellow et al., 2014) and Empirical Risk Minimization (ERM) (Vapnik, 2013) objectives, there has not been much focus on explicitly analyzing the nature of optimization in DAT. One direction of work aiming to improve generalization of ERM on unseen data focuses on developing algorithms that converge to a smooth (or a flat) minima (Foret et al., 2021; Keskar & Socher, 2017). However, we find that these techniques, when directly applied for DAT, do not significantly improve the generalization on the target domain (Sec. 4 and 7).

In this work, we analyze the loss landscape near the optimal point obtained by DAT, to gain insights into curvature. We first focus on the eigen-spectrum of Hessian of the task loss (ERM term for classification) where we find that using Stochastic Gradient Descent (SGD) as optimizer converges to a smoother minima in comparison to Adam (Kingma & Ba, 2014). Further we find that *smoother minima w.r.t. task loss leads to better generalization on the target domain*. Contrary to task loss, we find that smoothness enhancing formulation for adversarial components worsen performance, rendering ERM-based techniques which enhance smoothness for all loss components ineffective. Hence we introduce Smooth Domain Adversarial Training (SDAT), which aims only to reach a smooth minima w.r.t. task loss, and helps in generalizing better on the target domain. SDAT requires an additional gradient computation step and can be combined with existing methods with a few lines of code. We show the soundness of the SDAT method theoretically by proving a generalization bound (Sec. 4) on target error. We extensively verify the empirical efficacy of SDAT across various datasets for classification (i.e., DomainNet, VisDA-2017 and Office-Home), along with showing a prototypical application in DA for object detection, demonstrating it's diverse applicability. In summary, we make the following contributions:

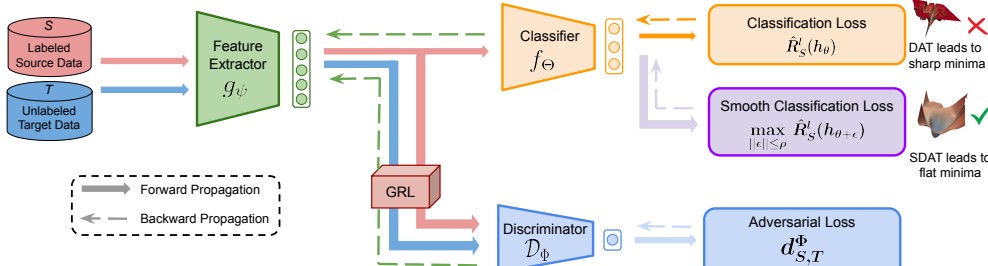

Figure 1: Overview of Smooth Domain Adversarial Training. Conventional approaches of smoothing loss do not discriminate between adversarial loss and task loss. Based on our theoretical analysis we propose SDAT which only focuses on smoothing task loss, leading to effective generalization on target domain. [1]

- We analyze the optimization procedure of DAT, establishing the correlation between the smoothness near optima w.r.t. task loss and generalization on the target domain.

- Contrary to ERM, we show through our theoretical and empirical analysis that smoothness enhancing adversarial formulation leads to sub-optimal performance.

- For enhancing the smoothness w.r.t. task loss near optima in DAT, we propose a novel and theoretically motivated SDAT that improves the generalization on the target domain. SDAT effectively increases the average performance of even state-of-the-art adversarial adaptation methods.

## 2 RELATED WORK

**Unsupervised Domain Adaptation**: It refers to a class of methods that aim to adapt models to work in a target domain distinct from what it was trained on. One of the most prominent lines of work is based on DAT (Ganin & Lempitsky, 2015). This involves using an additional discriminator to distinguish between samples of source and target domain. The goal of the model is to learn features that can not be distinguished between source and target. The follow-up works have improved this basic idea by introducing a class information based discriminator (CDAN (Long et al., 2018)), introducing a transferable normalization function (Wang et al., 2019) etc. In this work, we focus on analyzing and improving such methods. Another line of work involves DA by using self-training on target domain (Kundu et al., 2020b;a; Prabhu et al., 2020) which will not be the focus of this work.

**Smoothness of Loss Landscape:** As neural networks operate in the regime of over parameterized models, low error on training data does not always lead to better generalization (Keskar et al., 2017). Often it has been stated (He et al., 2019; Dziugaite & Roy, 2017) that smoother minima does generalize better on unseen data. But until recently, this was practically expensive as smoothing required additional costly computations. Recently, a method called Sharpness Aware Minimization (SAM) (Foret et al., 2021) has been proposed to find a smoother minima with an additional gradient computation step. SAM also improves the ImageNet model performance (Chen et al., 2021) on ImageNet-C and ImageNet-R (which are out of distribution). It has also been observed that smoothness w.r.t. input (image) is beneficial for domain adaptation (Shu et al., 2018; Cai et al., 2021), which motivates us to explore smoothness w.r.t weights (W) in case of DAT. However, the earlier work has focused on achieving a smoother minima w.r.t. W for ERM. Currently, no study has been done if the loss function is composed of both ERM and adversarial objectives (as present in DAT).

## 3 BACKGROUND

### 3.1 PRELIMINARIES

We will primarily focus on Unsupervised DA where we have labeled source data $S = \{(x_i^s, y_i^s)\}$ and unlabeled target data $T = \{(x_i^t)\}$. The source samples are assumed to be sampled i.i.d. from source

---

[1] Figures for the smooth minima and sharp minima are from (Foret et al., 2021) and used for illustration purposes only.

distribution $P_S$ defined on input space $\mathcal{X}$, similarly target samples are sampled i.i.d. from $P_T$. $\mathcal{Y}$ is used for denoting the label set which is $\{1, 2, \ldots, k\}$ in our case as we perform multiclass ($k$) classification. We denote $y : \mathcal{X} \to \mathcal{Y}$ a mapping from images to labels. Our task is to find a hypothesis function $h_\theta$ that has a low risk on the target distribution. The source risk (a.k.a expected error) of the hypothesis $h_\theta$ is defined with respect to loss function $l$ as: $R_S^l(h_\theta) = \mathbb{E}_{x \sim P_S}[l(h_\theta(x), y(x))]$. The target risk $R_T^l(h_\theta)$ is defined analogously. The empirical versions of source and target risk will be denoted by $\hat{R}_S^l(h_\theta)$ and $\hat{R}_T^l(h_\theta)$. All notations used in paper are summarized in Table F. In this work we build on the domain adaption theory of (Acuna et al., 2021) which is a generalization of Ben-David et al. (2010). We first define the discrepancy between the two domains.

**Definition 3.1** ($D_{h_\theta, \mathcal{H}}^\phi$ discrepancy). *The discrepancy between two domains $P_S$ and $P_T$ is defined as following:*

$$D_{h_\theta, \mathcal{H}}^\phi(P_S || P_T) := \sup_{h' \in \mathcal{H}} [\mathbb{E}_{x \sim P_S}[l(h_\theta(x), h'(x))]] - [\mathbb{E}_{x \sim P_T}[\phi^*(l(h_\theta(x), h'(x)))]] \tag{1}$$

*Here $\phi^*$ is a frenchel conjugate of a lower semi-continuous convex function $\phi$ that satisfies $\phi(1) = 0$, and $\mathcal{H}$ is the set of all possible hypothesis (i.e. Hypothesis Space).*

This discrepancy distance $D_{h_\theta, \mathcal{H}}^\phi$ is based on variational formulation of f-divergence (Nguyen et al., 2010) for the convex function $\phi$. The $D_{h_\theta, \mathcal{H}}^\phi$ is the lower bound estimate of the f-divergence function $D^\phi(P_S || P_T)$. See Lemma 4 in (Acuna et al., 2021) for additional details. We state a bound on target risk $R_T^l(h_\theta)$ based on $\mathcal{D}_{h_\theta, \mathcal{H}}^\phi$ discrepancy (Acuna et al., 2021):

**Theorem 1** (**Generalization bound**). *Suppose $l : \mathcal{Y} \times \mathcal{Y} \to [0, 1] \subset dom\ \phi^*$. Let $h^*$ be the ideal joint classifier with least $\lambda^* = R_S^l(h^*) + R_T^l(h^*)$ (i.e. joint risk) in $\mathcal{H}$. We have the following relation between source and target risk:*

$$R_T^l(h_\theta) \leq R_S^l(h_\theta) + D_{h_\theta, \mathcal{H}}^\phi(P_S || P_T) + \lambda^* \tag{2}$$

The above generalization bound shows that the target risk $R_T^l(h_\theta)$ is upper bounded by the source risk $R_S^l(h_\theta)$ and the discrepancy term $D_{h_\theta, \mathcal{H}}^\phi$ along with an irreducible constant error $\lambda^*$. Hence, this infers that reducing source risk and discrepancy lead a to reduction in target risk. Based on this, we concretely define the unsupervised adversarial adaptation procedure in the next section.

## 3.2 Unsupervised Domain Adaptation

In this section we first define the components of the framework we use for our purpose: $h_\theta = f_\Theta \circ g_\psi$ where $g_\psi$ is the feature extractor and $f_\Theta$ is the classifier. The domain discriminator $\mathcal{D}_\Phi$, used for estimating the discrepancy between $P_S$ and $P_T$ is a classifier whose goal is to distinguish between the features of two domains. For minimizing the target risk (Th. 1), the optimization problem can be written as:

$$\min_\theta \mathbb{E}_{x \sim P_S}[l(h_\theta(x), y(x))] + D_{h_\theta, \mathcal{H}}^\phi(P_S || P_T) \tag{3}$$

The discrepancy term under some assumptions (refer App. B) can be upper bounded by a tractable term:

$$D_{h_\theta, \mathcal{H}}^\phi(P_S || P_T) \leq \max_\Phi d_{S,T}^\Phi \tag{4}$$

where $d_{S,T}^\Phi = \mathbb{E}_{x \sim P_S}[\log(\mathcal{D}_\Phi(g_\psi(x)))] + \mathbb{E}_{x \sim P_T} \log[1 - \mathcal{D}_\Phi(g_\psi(x))]$. This leads to the final optimization objective of:

$$\min_\theta \max_\Phi \mathbb{E}_{x \sim P_S}[l(h_\theta(x), y(x))] + d_{S,T}^\Phi \tag{5}$$

The first term in practice is empirically approximated by using finite samples $\hat{R}_S^l(h_\theta)$ and used as task loss (classification) for minimization. The empirical estimate of the second term is adversarial loss which is optimized using a gradient reversal layer (GRL) as it has a min-max form. (Overview in Fig. 1) The above procedure composes DAT, and we use CDAN (Long et al., 2018) as our default DAT method.

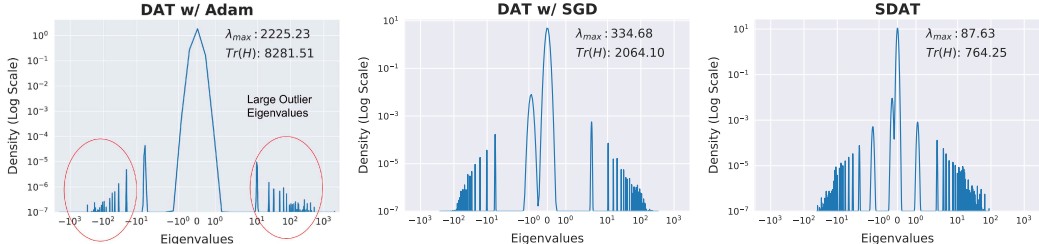

Figure 2: Eigen Spectral Density plots of Hessian ($\nabla^2 \hat{R}_S^l(h_\theta)$) for Adam (left), SGD (middle) and SDAT (right) on Art → Clipart. Each plot contains the maximum eigenvalue ($\lambda_{max}$) and the trace of the Hessian ($Tr(H)$), which are indicators of the smoothness (Lower $Tr(H)$ and $\lambda_{max}$ indicate the presence of smoother loss surface). Low range of eigenvalues (x-axis), $Tr(H)$ and $\lambda_{max}$ for SGD indicates that it reaches a smoother minima compared to Adam. SDAT reaches a smoother minima compared to DAT with either SGD and Adam.

## 4 ANALYSIS OF SMOOTHNESS

In this section, we analyze the curvature properties of the loss with respect to the parameters. Specifically, we focus on analyzing the Hessian of empirical source risk $H = \nabla_\theta^2 \hat{R}_S^l(h_\theta)$ which is the Hessian of classification (task) loss term. For quantifying the smoothness, we measure the trace $Tr(H)$ and maximum eigenvalue of Hessian ($\lambda_{max}$) as a proxy for quantifying smoothness. This is motivated by analysis of which states that the high value of $\lambda_{max}$ and $Tr(H)$ are indicative of low smoothness (Jastrzebski et al., 2020). We articulate our conjecture informally below:

**Conjecture 1.** *Smoothing of empirical source risk (i.e. task loss) $\hat{R}_S^l(h_\theta)$ leads to efficient DAT. In other words, decreasing $\lambda_{max}$ of $\nabla_\theta^2 \hat{R}_S^l(h_\theta)$ leads to reduced error on target domain $\hat{R}_T^l(h_\theta)$.*

For verifying our conjecture, we analyze the eigen spectrum of the Hessian $\hat{R}_T^l(h_\theta)$ where we find that in contrast to standard ERM (Ghorbani et al., 2019) the negative eigenvalues do not disappear as the training progresses. We show the $\lambda_{max}, Tr(H)$ and eigen spectrum for different algorithms, namely DAT w/ Adam, DAT w/ SGD and our proposed SDAT (which is described in detail in later sections) in Fig. 2. We find that *high smoothness leads to better generalization on the target domain*. We also provide additional results in Fig. 3 for empirical verification of the conjecture. Our conjecture also explains the reason for widespread usage of SGD for DAT as SGD converges to smoother minima (Ganin & Lempitsky, 2015; Long et al., 2018; Saito et al., 2018a) which leads to efficient DAT, even though Adam has shown to be effective for min-max optimization (Gemp & McWilliams, 2019). More details regarding the Hessian analysis are provided in App. D.

### 4.1 SMOOTHING LOSS LANDSCAPE

In this section we first introduce the losses which are based on Sharpness Aware Minimization (Foret et al., 2021) (SAM). The basic idea of SAM is to find a smoother minima (i.e. low loss in $\epsilon$ neighborhood of $\theta$) by using the following objective given formally below:

$$\min_\theta \max_{||\epsilon|| \le \rho} L_{obj}(\theta + \epsilon) \tag{6}$$

Here $L_{obj}$ is any objective function to be minimized and $\rho \ge 0$ is a hyperparameter which defines the maximum norm of the $\epsilon$. Since finding the exact solution of inner maximization is hard, SAM maximizes the first order approximation:

$$\hat{\epsilon}(\theta) \approx \arg\max_{||\epsilon|| \le \rho} L_{obj}(\theta) + \epsilon^T \nabla_\theta L_{obj}(\theta) = \rho \nabla_\theta L_{obj}(\theta) / ||\nabla_\theta L_{obj}(\theta)||_2 \tag{7}$$

The $\hat{\epsilon}(\theta)$ is added to the weights $\theta$. The gradient update for $\theta$ is then computed as $\nabla_\theta L_{obj}(\theta)|_{\theta + \hat{\epsilon}(\theta)}$. The above procedure can be seen as a generic smoothness enhancing formulation for any $L_{obj}$. We now analogously introduce the sharpness aware source risk for finding a smooth minima:

$$\max_{||\epsilon|| \le \rho} R_S^l(h_{\theta + \epsilon}) = \max_{||\epsilon|| \le \rho} \mathbb{E}_{x \sim P_S}[\, l(h_{\theta + \epsilon}(x), f(x))] \tag{8}$$

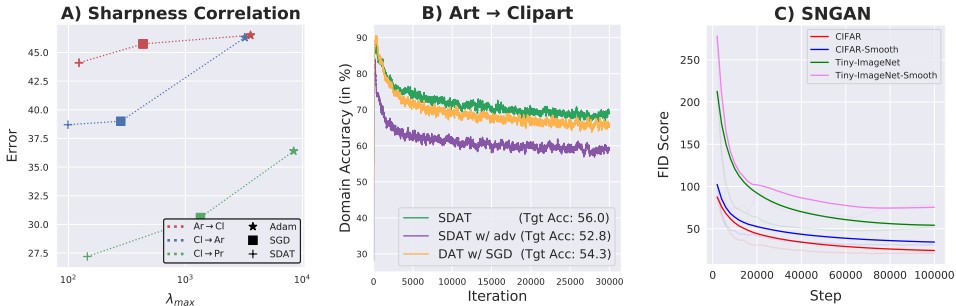

Figure 3: A) Error on Target Domain (y-axis) for Office-Home dataset against maximum eigenvalue $\lambda_{max}$ of classification loss in DAT. When compared to SGD, Adam converges to a non-smooth minima (high $\lambda_{max}$), leading to a high error on target. B) Domain Accuracy (vs iterations) is lower when discriminator is smooth (i.e. SDAT w/ adv), which indicates suboptimal discrepancy estimation $d_{s,t}^{\Phi}$ C) SNGAN performance on different datasets, smoothing discriminator in GAN also leads to inferior GAN performance (higher FID) across both datasets.

We also now define the sharpness aware discrepancy estimation objective below:

$$\max_{\Phi} \min_{||\epsilon||\leq\rho} d_{S,T}^{\Phi+\epsilon} \tag{9}$$

As $d_{S,T}^{\Phi}$ is to be maximized the sharpness aware objective will have $\min_{||\epsilon||\leq\rho}$ instead of $\max_{||\epsilon||\leq\rho}$, as it needs to find smoother maxima. We now theoretically analyse the difference in discrepancy estimation for smooth version $d_{S,T}^{\Phi''}$ (Eq. 9) in comparison to non-smooth version $d_{S,T}^{\Phi'}$ (Eq. 4). Assuming $\mathcal{D}_{\Phi}$ is a $L$-smooth (common assumption for non-convex optimization (Carmon et al., 2020)), $\eta$ is a small constant and $d_{S,T}^{*}$ the optimal discrepancy, the theorem states:

**Theorem 2.** *For a given classifier $h_{\theta}$ and one step of (steepest) gradient ascent i.e.* $\Phi' = \Phi + \eta(\nabla d_{S,T}^{\Phi}/||\nabla d_{S,T}^{\Phi}||)$ *and* $\Phi'' = \Phi + \eta(\nabla d_{S,T}^{\Phi}|_{\Phi+\hat{\epsilon}(\Phi)}/||\nabla d_{S,T}^{\Phi}|_{\Phi+\hat{\epsilon}(\Phi)}||)$

$$d_{S,T}^{\Phi'} - d_{S,T}^{\Phi''} \leq \eta(1-\cos\alpha)\sqrt{2L(d_{S,T}^{*} - d_{S,T}^{\Phi})} \tag{10}$$

*where $\alpha$ is the angle between $\nabla d_{S,T}^{\Phi}$ and $\nabla d_{S,T}^{\Phi}|_{\Phi+\hat{\epsilon}(\Phi)}$.*

The $d_{S,T}^{\Phi'}$ (non-smooth version) can exceed $d_{S,T}^{\Phi''}$ (smooth discrepancy) significantly, as the term $d_{S,T}^{*} - d_{S,T}^{\Phi} \not\to 0$, as the $h_{\theta}$ objective is to oppose the convergence of $d_{S,T}^{\Phi}$ to optima $d_{S,T}^{*}$ (min-max training in Eq. 11). Thus $d_{S,T}^{\Phi'}$ can be a better estimate of discrepancy in comparison to $d_{S,T}^{\Phi''}$. A better estimate of $d_{s,t}^{\Phi}$ helps in effectively reducing the discrepancy between $P_S$ and $P_T$, hence leads to reduced $R_T^l(h_{\theta})$. This is also observed in practice that smoothing the discriminator (SDAT w/ adv in Fig. 3) leads to low domain classification accuracy (proxy measure for $d_{s,t}^{\Phi}$) in comparison to DAT. Due to ineffective discrepancy estimation, SDAT w/ adv results in sub-optimal generalization on target domain i.e. high target error $R_T^l(h_{\theta})$ (Fig. 3). For further establishing the generality of sub-optimality of smooth adversarial loss, we also perform experiments on Spectral Normalised Generative Adversarial Networks (SNGAN) (Miyato et al., 2018). In case of SNGAN we also find that smoothing discriminator through SAM leads to suboptimal performance (higher FID) as in Fig. 3. The above evidences indicates that *smoothing the adversarial loss leads to sub-optimality*, hence it should not be done in practice. The proof of the above theorem and additional experimental details are provided in the supplementary (refer App. C and App. E).

## 4.2 SMOOTH DOMAIN ADVERSARIAL TRAINING (SDAT)

We propose smooth domain adversarial training which only focuses on converging to smooth minima w.r.t. task loss (i.e. empirical source risk), whereas does no change for the discrepancy term. We define the optimization objective of our smooth domain adversarial training below:

$$\min_{\theta} \max_{\Phi} \max_{||\epsilon|| \leq \rho} \mathbb{E}_{x \sim P_S}[l(h_{\theta+\epsilon}(x), y(x))] + d_{S,T}^{\Phi} \tag{11}$$

The first term is the sharpness aware risk, and the second term is the discrepancy term which is not smooth in our procedure. The term $d_{S,T}^{\Phi}$ estimates $D_{h_\theta,H}^{\phi}(P_S||P_T)$ discrepancy. We empirically find that this optimization procedure effectively reduces the generalization error on the target domain compared to all other alternatives. We now show that optimizing Eq. 11 reduces $R_T^l(h_\theta)$ through a generalization bound. This bound establishes that our procedure is also consistent (i.e. in case of infinite data the upper bound is tight), similar to the DAT (Ganin et al., 2016) baseline.

**Theorem 3.** *Suppose $l$ is the loss function, we denote $\lambda^* := R_S^l(h^*) + R_T^l(h^*)$ and let $h^*$ be the ideal joint hypothesis:*

$$R_T^l(h_\theta) \leq \max_{||\epsilon|| \leq \rho} \hat{R}_S^l(h_{\theta+\epsilon}) + D_{h_\theta,H}^{\phi}(P_S||P_T) + \gamma(||\theta||_2^2/\rho^2) + \lambda^*. \tag{12}$$

*where $\gamma : \mathbb{R}^+ \to \mathbb{R}^+$ is a strictly increasing function.*

The bound is similar to generalization bounds for domain adaptation (Ben-David et al., 2010; Acuna et al., 2021). The main difference is the sharpness aware risk term $\max_{||\epsilon|| \leq \rho} \hat{R}_S^l(h_\theta)$ in place of source risk $R_S^l(h_\theta)$, and an additional term that depends on the norm of the weights $\gamma(||\theta||_2^2/\rho^2)$. The first is minimized by decreasing the empirical sharpness aware source risk by using SAM loss shown in Sec. 4. The second term is reduced by decreasing the discrepancy between source and target domains. The third term, as it is a function of norm of weights $||\theta||_2^2$, can be reduced by using either L2 regularization or weight decay. Since we assume that the $\mathcal{H}$ hypothesis class we have is rich, the $\lambda^*$ term is small. We now show the improvements due to SDAT empirically in the following sections.

## 5 ADAPTATION FOR CLASSIFICATION

We evaluate our proposed method on three datasets: Office-Home, VisDA-2017, and DomainNet, as well as by combining SDAT with two DAT based DA techniques: CDAN and CDAN+MCC.

### 5.1 DATASETS

**Office-Home** (Venkateswara et al., 2017): Office-Home consists of around 15,500 images from 65 classes and four distinct domains: Art (Ar), Clipart (Cl), Product (Pr) and Real World (Rw).
**VisDA-2017** (Peng et al., 2017): VisDA is a dataset that focuses on the transition from simulation to real world and contains approximately 280K images across 12 classes.
**DomainNet** (Peng et al., 2019): DomainNet consists of 0.6 million images across 345 classes belonging to six domains. The domains are infograph (inf), clipart (clp), painting (pnt), sketch (skt), real and quickdraw.

### 5.2 DOMAIN ADAPTATION METHODS

**CDAN** (Long et al., 2018): Conditional Domain Adversarial network is a popular DA algorithm that improves the performance of the DANN algorithm. CDAN introduces the idea of multi-linear conditioning to align the source and target distributions better. CDAN* in Table 1 and 4 refers to our implementation of CDAN method.
**CDAN + MCC** (Jin et al., 2020): In this method, the minimum class confusion loss term is added as a regularizer to CDAN. Minimum class confusion is a non-adversarial term that minimizes the pairwise class confusion on the target domain. This achieves state of the art accuracy among adversarial adaptation methods.

### 5.3 IMPLEMENTATION DETAILS

We implement our proposed method in the Transfer-Learning-Library (Junguang Jiang & Long, 2020) toolkit developed in PyTorch (Paszke et al., 2019). The main difference between the performance reported in the CDAN and our implementation (CDAN*) is the batch normalization layer in the domain classifier, which enhances performance.

Table 1: Accuracy (%) on Office-Home for unsupervised domain adaptation (ResNet-50). CDAN+MCC w/ SDAT outperforms other sophisticated state-of-the-art DA techniques. CDAN w/ SDAT improves over performance of CDAN by 1.1%.

| Method | Ar→Cl | Ar→Pr | Ar→Rw | Cl→Ar | Cl→Pr | Cl→Rw | Pr→Ar | Pr→Cl | Pr→Rw | Rw→Ar | Rw→Cl | Rw→Pr | Avg |
|---|---|---|---|---|---|---|---|---|---|---|---|---|---|
| ResNet-50 (He et al., 2016) | 34.9 | 50.0 | 58.0 | 37.4 | 41.9 | 46.2 | 38.5 | 31.2 | 60.4 | 53.9 | 41.2 | 59.9 | 46.1 |
| DAN (Long et al., 2015) | 43.6 | 57.0 | 67.9 | 45.8 | 56.5 | 60.4 | 44.0 | 43.6 | 67.7 | 63.1 | 51.5 | 74.3 | 56.3 |
| DANN (Ganin et al., 2016) | 45.6 | 59.3 | 70.1 | 47.0 | 58.5 | 60.9 | 46.1 | 43.7 | 68.5 | 63.2 | 51.8 | 76.8 | 57.6 |
| JAN (Long et al., 2017) | 45.9 | 61.2 | 68.9 | 50.4 | 59.7 | 61.0 | 45.8 | 43.4 | 70.3 | 63.9 | 52.4 | 76.8 | 58.3 |
| CDAN (Long et al., 2018) | 49.0 | 69.3 | 74.5 | 54.4 | 66.0 | 68.4 | 55.6 | 48.3 | 75.9 | 68.4 | 55.4 | 80.5 | 63.8 |
| MDD (Zhang et al., 2019) | 54.9 | 73.7 | 77.8 | 60.0 | 71.4 | 71.8 | 61.2 | 53.6 | 78.1 | 72.5 | 60.2 | 82.3 | 68.1 |
| f-DAL-pearson + alignment (Acuna et al., 2021) | 56.7 | 77.0 | 81.1 | 63.1 | 72.2 | 75.9 | 64.5 | 54.4 | 81.0 | 72.3 | 58.4 | 83.7 | 70.0 |
| SRDC (Tang et al., 2020) | 52.3 | 76.3 | 81.0 | **69.5** | 76.2 | 78.0 | 68.7 | 53.8 | 81.7 | 76.3 | 57.1 | 85.0 | 71.3 |
| CDAN*[2] | 54.3 | 70.6 | 76.8 | 61.3 | 69.5 | 71.3 | 61.7 | 55.3 | 80.5 | 74.8 | 60.1 | 84.2 | 68.4 |
| CDAN w/ SDAT | 56.0 | 72.2 | 78.6 | 62.5 | 73.2 | 71.8 | 62.1 | 55.9 | 80.3 | 75.0 | 61.4 | 84.5 | 69.5 |
| CDAN + MCC (Jin et al., 2020) | 57.0 | 76.0 | 81.6 | 64.9 | 75.9 | 75.4 | 63.7 | 56.1 | 81.2 | 74.2 | 63.9 | 85.4 | 71.3 |
| CDAN + MCC w/ SDAT | **58.2** | **77.1** | **82.2** | 66.3 | **77.6** | 76.8 | 63.3 | **57.0** | **82.2** | 74.9 | **64.7** | **86.0** | **72.2** |

We use a ResNet-50 backbone for Office-Home experiments and a ResNet-101 backbone for VisDA-2017 and DomainNet experiments. The backbone is initialized with ImageNet weights. We use a learning rate of 0.01 with batch size 32 in all of our experiments. We tune $\rho$ value in SDAT for a particular dataset and use the same value across domains. The $\rho$ value is set to 0.02 for the Office-Home experiments, 0.005 for the VisDA-2017 experiments and 0.05 for the DomainNet experiments. More details are present in supplementary (refer App. F).

## 5.4 RESULTS

Table 2 shows the results on the large and challenging DomainNet dataset across five domains as done in (Junguang Jiang & Long, 2020). The proposed method improves the performance of CDAN significantly across all source-target pairs. On specific source-target pairs like info-graph → real, the performance increase is 4.5%. The overall performance of CDAN is improved by nearly 1.8% which is significant considering the large number of classes and images present in DomainNet.

For the Office-Home dataset, we compare our methods with other domain adaptation algorithms including DANN, SRDC, MDD and f-DAL. The results for the Office-Home dataset are shown in Table 1. We can see that adding SDAT improves the performance on both CDAN and CDAN+MCC across all the transfer tasks. CDAN+MCC w/ SDAT achieves state-of-the-art adversarial adaptation performance on the Office-Home dataset.

The class-wise accuracy on VisDA-2017 are reported in Table 4. CDAN w/ SDAT improves the overall performance of CDAN by more than 1.5%. CDAN w/ SDAT improves the performance of underperforming minority classes like bicycle and car. Additional baselines and results are reported in supplementary (refer App. G) along with a discussion on statistical significance (App. J) .

Table 2: Results on DomainNet with CDAN w/ SDAT. The number in the parenthesis refers to the increase in accuracy with respect to CDAN.

| Target (→) Source (↓) | clp | inf | pnt | real | skt | Avg |
|---|---|---|---|---|---|---|
| clp | - | 22.0 (+1.4) | 41.5 (+2.6) | 57.5 (+1.5) | 47.2 (+2.3) | 42.1 (+2.0) |
| inf | 33.9 (+2.3) | - | 30.3 (+1.0) | 48.1 (+4.5) | 27.9 (1.5) | 35.0 (+2.3) |
| pnt | 47.5 (+3.4) | 20.7 (+0.9) | - | 58.0 (+0.8) | 41.8 (+1.8) | 42.0 (+1.7) |
| real | 56.7 (+0.9) | 25.1 (+0.7) | 53.6 (+0.4) | - | 43.9 (+1.6) | 44.8 (+1.0) |
| skt | 58.7 (+2.7) | 21.8 (+1.1) | 48.1 (+2.8) | 57.1 (+2.2) | - | 46.4 (+2.2) |
| Avg | 49.2 (+2.3) | 22.4 (+1.0) | 43.4 (+1.7) | 55.2 (+2.2) | 40.2 (+1.8) | 42.1 (+1.8) |

## 6 ADAPTATION FOR OBJECT DETECTION

To further validate our approach's generality and extensibility, we did experiments on DA for object detection. We use the same setting as proposed in DA-Faster (Chen et al., 2018) with all domain adaptation components and use it as our baseline. We use the mean Average Precision at 0.5 IoU (mAP) as our evaluation metric. In object detection, the smoothness enhancement can be achieved in two ways (empirical comparison in Sec. 6.2) :

**a) DA-Faster w/ SDAT-Classification:** Smoothness enhancement for classification loss.
**b) DA-Faster w/ SDAT:** Smoothness enhancment for the combined classification and regression loss.

Table 4: Accuracy (%) on VisDA-2017 for unsupervised domain adaptation (ResNet-101). The **mean** column contains mean across all classes. SDAT particularly improves the accuracy in classes that have comparatively low CDAN performance.

| Method | plane | bcybl | bus | car | horse | knife | mcyle | persn | plant | sktb | train | truck | mean |
|---|---|---|---|---|---|---|---|---|---|---|---|---|---|
| ResNet (He et al., 2016) | 55.1 | 53.3 | 61.9 | 59.1 | 80.6 | 17.9 | 79.7 | 31.2 | 81.0 | 26.5 | 73.5 | 8.5 | 52.4 |
| DANN (Ganin et al., 2016) | 81.9 | 77.7 | 82.8 | 44.3 | 81.2 | 29.5 | 65.1 | 28.6 | 51.9 | 54.6 | 82.8 | 7.8 | 57.4 |
| DAN (Long et al., 2015) | 87.1 | 63.0 | 76.5 | 42.0 | 90.3 | 42.9 | 85.9 | 53.1 | 49.7 | 36.3 | 85.8 | 20.7 | 61.1 |
| MCD (Saito et al., 2018b) | 87.0 | 60.9 | 83.7 | 64.0 | 88.9 | 79.6 | 84.7 | 76.9 | 88.6 | 40.3 | 83.0 | 25.8 | 71.9 |
| CDAN (Long et al., 2018) | 85.2 | 66.9 | 83.0 | 50.8 | 84.2 | 74.9 | 88.1 | 74.5 | 83.4 | 76.0 | 81.9 | 38.0 | 73.9 |
| AFN (Xu et al., 2019) | 93.6 | 61.3 | **84.1** | 70.6 | 94.1 | 79.0 | **91.8** | 79.6 | 89.9 | 55.6 | **89.0** | 24.4 | 76.1 |
| MCC (Jin et al., 2020) | 88.1 | 80.3 | 80.5 | **71.5** | 90.1 | 93.2 | 85.0 | 71.6 | 89.4 | 73.8 | 85.0 | 36.9 | 78.8 |
| CDAN*[2] | 94.9 | 72.0 | 83.0 | 57.3 | 91.6 | 95.2 | 91.6 | 79.5 | 85.8 | 88.8 | 87.0 | 40.5 | 80.6 |
| CDAN w/ SDAT | 94.8 | 77.1 | 82.8 | 60.9 | 92.3 | 95.2 | 91.7 | **79.9** | 89.9 | 91.2 | 88.5 | 41.2 | 82.1 |
| CDAN+MCC (Jin et al., 2020) | 95.0 | 84.2 | 75.0 | 66.9 | **94.4** | 97.1 | 90.5 | 79.8 | 89.4 | 89.5 | 86.9 | 54.4 | 83.6 |
| CDAN+MCC w/ SDAT | **95.8** | **85.5** | 76.9 | 69.0 | 93.5 | **97.4** | 88.5 | 78.2 | **93.1** | **91.6** | 86.3 | **55.3** | **84.3** |

## 6.1 EXPERIMENTAL SETUP

We evaluate our proposed approach on object detection on two different domain shifts:

**Pascal to Clipart** ($P \rightarrow C$): Pascal (Everingham et al., 2010) is a real-world image dataset which consists images with 20 different object categories. Clipart (Inoue et al., 2018) is a graphical image dataset with complex backgrounds and has the same 20 categories as Pascal. We use Resnet-101 (He et al., 2016) backbone for Faster R-CNN (Ren et al., 2015) following Saito et al. (2019).
**Cityscapes to Foggy Cityscapes** ($C \rightarrow Fc$): Cityscapes (Cordts et al., 2016) is a street scene dataset for driving, whose images are collected in clear weather. Foggy Cityscapes (Sakaridis et al., 2018) dataset is synthesized from Cityscapes for the foggy weather. We use Resnet-50 (He et al., 2016) as the backbone for Faster R-CNN for experiments on this task. Both domains have the same 8 object categories with instance labels.

The training is done via SGD with momentum 0.9 for 70k iterations with the learning rate of $10^{-3}$, and then dropped to $10^{-4}$ after 50k iterations. We split the target data into train and validation sets and report the best mAP on validation data. Additional experimental and implementation details are present in supplementary (refer App. F).

## 6.2 RESULTS

Table 3 shows the results on two domain shifts with varying batch size (*bs*) during training. We find that only smoothing w.r.t. classification loss is much more effective (SDAT-Classification) than smoothing w.r.t. combined classification and regression loss (SDAT). On average, SDAT-Classification produces an mAP gain of 2.0% compared to SDAT, and 2.8% compared to DA-Faster baseline.

The proposed SDAT-Classification significantly outperforms DA-Faster baseline and improves mAP by 1.3% on $P \rightarrow C$ and by 2.8% on $C \rightarrow Fc$. It is noteworthy that increase in performance of SDAT-Classification is consistent even after training with higher batch size ($bs = 8$) achieving improvement of 4.3% in mAP.

Table 3: Results on DA for object detection.

| Method | $C \rightarrow Fc$ (bs=2) | $P \rightarrow C$ (bs=2) | $P \rightarrow C$ (bs=8) |
|---|---|---|---|
| DA-Faster (Chen et al., 2018) | 35.21 | 29.96 | 26.40 |
| DA-Faster w/ SDAT | 37.47 | 29.04 | 27.64 |
| DA-Faster w/ SDAT-Classification | **38.00** | **31.23** | **30.74** |

Table 3 also shows that even DA-Faster w/ SDAT (i.e. smoothing both classification and regression) outperforms DA-Faster by 0.9 % on average across all experiments. The improvement due to SDAT on adaptation for object detection shows the generality of SDAT across techniques that have some form of adversarial component present in the loss formulation.

## 7 DISCUSSION

**How much smoothing is optimal?**: Figure 4 (A) shows the ablation on $\rho$ value (higher $\rho$ value corresponds to more smoothing) on the Ar→Cl and Cl→Pr from Office-Home dataset with CDAN backbone. The performance of the different values of $\rho$ is higher than the baseline with $\rho = 0$. It

---

[2]Our Implementation of CDAN. Refer to Section 5.3.

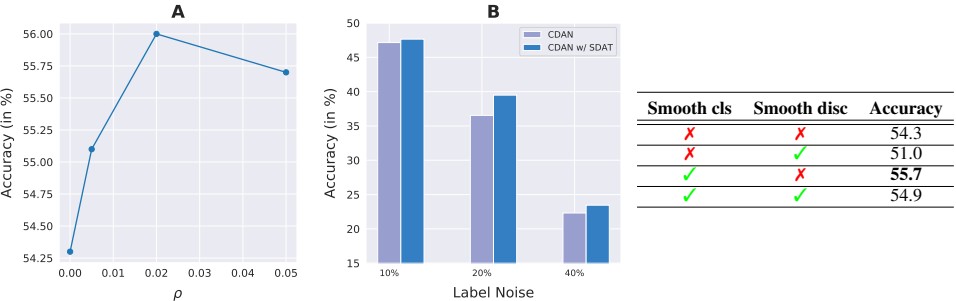

Figure 4: Analysis of SDAT for Ar → Cl split of Office-Home dataset. A) Variation of target accuracy with maximum perturbation $\rho$. B) Comparison of accuracy of SDAT with DAT for different ratio of label noise. C) Comparison of accuracy when smoothing is applied to various loss components.

can be seen that $\rho = 0.02$ works best among all the different values and outperforms the baseline by at least 1.5%. We found that the same $\rho$ value usually worked well across domains in a dataset, but different $\rho$ was optimal for different datasets.

**Which components benefit from smooth optima?**: Figure 4 (C) shows the effect of introducing smoothness enhancement for different components in DAT. For this we use SAM on a) classifier (SDAT) b) discriminator (SDAT w/ adv) c) both classifier and discriminator (SDAT-all). It can be seen that smoothing the adversarial component (SDAT w/ adv) reduces the performance to 51.0%, which is significantly lower than even the DAT baseline.

**Is it Robust to Label Noise?**: In practical, real-world scenarios, the labeled datasets are often corrupted with some amount of label noise. Due to this, performing domain adaptation with such data is challenging. We find that smoother minima through SDAT lead to robust models which generalize well on the target domain. Figure 4 (B) provides the comparison of SGD vs. SDAT for different percentages of label noise injected into training data (by flipping the labels).

**Is it better than other smoothing techniques?** To answer this question, we compare SDAT with different smoothing techniques originally proposed for ERM. We specifically compare our method against DAT, Label Smoothing (LS) (Szegedy et al., 2016), and VAT (Miyato et al., 2019). Stutz et al. (2021) recently showed that these techniques produce a significantly smooth loss landscape in comparison to SGD. We also compare with a very recent

Table 5: Performance comparison across different loss smoothing techniques on Office-Home. SDAT outperforms other smoothing techniques in each case consistently.

| Method | Ar→Cl | Cl→Pr | Rw→Cl | Pr→Cl | Avg | |
|---|---|---|---|---|---|---|
| DAT | 54.3 | 69.5 | 60.1 | 55.3 | 59.2 | |
| VAT | 54.6 | 70.7 | 60.8 | 54.4 | 60.1 | (+0.9) |
| SWAD | 54.6 | 71.0 | 60.9 | 55.2 | 60.4 | (+1.2) |
| LS | 53.6 | 71.6 | 59.9 | 53.4 | 59.6 | (+0.4) |
| SDAT | **56.0** | **73.2** | **61.4** | **55.9** | **61.6** | (+2.4) |

SWAD (Cha et al., 2021) technique which is shown effective for domain generalization. For this, we run our experiments on four different splits of the Office-Home dataset and summarize our results in Table 5. We find that techniques for ERM (LS and VAT) fail to provide significant consistent gain in performance which also confirms the requirement of specific smoothing strategies for DAT. We find that SDAT even outperforms SWAD on average by a significant margin of 1.2%. Additional details regarding the specific methods are provided in the supplementary (refer App. H).

## 8   CONCLUSION

In this work, we analyse the curvature of loss surface of DAT used extensively for Unsupervised Domain Adaptation. We find that converging to a smooth minima w.r.t. task loss (i.e., empirical source risk) leads to better generalization on the target domain. We also theoretically and empirically show that smoothness enhancing for adversarial components of loss lead to sub-optimal results, hence should be avoided in practice. We then introduce our practical and effective method, SDAT, which only increases the smoothness w.r.t. task loss, leading to better generalization on the target domain. SDAT leads to an effective increase even for the state of the art methods for adversarial domain adaptation and can be incorporated with just a few lines of code change. One limitation of SDAT is presence of no automatic way of selecting $\rho$ (determines extent of smoothness) which is a good future direction to explore.

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

APPENDICES

## A    NOTATION TABLE

Table S1 contains all the notations used in the paper and the proofs of theorems.

Table S1: The notations used in the paper and the corresponding meaning.

| Notation | Meaning |
|---|---|
| $S$ | Labeled Source Data |
| $T$ | Unlabelled Target Data |
| $P_S$ (or $P_T$) | Source (or Target) Distribution |
| $\mathcal{X}$ | Input space |
| $\mathcal{Y}$ | Label space |
| $y(\cdot)$ | Maps image to labels |
| $h_\theta$ | Hypothesis function |
| $R_S^l(h_\theta)$ (or $R_T^l(h_\theta)$) | Source (or Target) risk |
| $\hat{R}_S^l(h_\theta)$ (or $\hat{R}_T^l(h_\theta)$) | Empirical Source (or Target) risk |
| $\mathcal{H}$ | Hypothesis space |
| $D_{h_\theta,\mathcal{H}}^\phi(P_S||P_T)$ | Discrepancy between two domains $P_S$ and $P_T$ |
| $g_\psi$ | Feature extractor |
| $f_\Theta$ | Classifier |
| $\mathcal{D}_\Phi$ | Domain Discriminator |
| $d_{S,T}^\Phi$ | Tractable Discrepancy Estimate |
| $\nabla_\theta^2 \hat{R}_S^l(h_\theta)$ (or $H$) | Hessian of classification loss |
| $Tr(H)$ | Trace of Hessian |
| $\lambda_{max}$ | Maximum eigenvalue of Hessian |
| $\epsilon$ | Perturbation |
| $\rho$ | Maximum norm of $\epsilon$ |

## B    CONNECTION OF DISCREPANCY TO $d_{S,T}^\Phi$ (EQ. 4) IN MAIN PAPER

We refer reader to Appendix C.2 of Acuna et al. (2021) for relation of $d_{S,T}^\Phi$. The $d_{S,T}^\Phi$ term defined in Eq. 4 given as:

$$d_{S,T}^\Phi = \mathbb{E}_{x \sim P_S}[\log(\mathcal{D}_\Phi(g_\psi(x)))] + \mathbb{E}_{x \sim P_T}[\log(1 - \mathcal{D}_\Phi(g_\psi(x)))] \tag{S1}$$

The above term is exactly the Eq. C.1 in Acuna et al. (2021) where they show that optimal $d_{S,T}^\Phi$ i.e.:

$$\max_\Phi d_{S,T}^\Phi = D_{JS}(P_S||P_T) - 2\log(2) \tag{S2}$$

Hence we can say from result in Eq. 4 is a consequence of Lemma 1 and Proposition 1 in (Acuna et al., 2021), assuming that $D_\Phi$ satisfies the constraints in Proposition 1.

## C    PROOF OF THEOREMS

In this section we provide proofs for the theoretical results present in the paper:

**Theorem 1 (Generalization bound).** *Suppose $l : \mathcal{Y} \times \mathcal{Y} \to [0,1] \subset dom\ \phi^*$. Let $h^*$ be the ideal joint classifier with error $\lambda^* = R_S^l(h^*) + R_T^l(h^*)$. We have the following relation between source and target risk:*

$$R_T^l(h_\theta) \leq R_S^l(h_\theta) + D_{h_\theta,\mathcal{H}}^\phi(P_S||P_T) + \lambda^* \tag{S3}$$

*Proof.* We refer the reader to Theorem 2 in Appendix B of Acuna et al. (2021) for the detailed proof the theorem. □

We now introduce a Lemma for smooth functions which we will use in the proofs subsequently:

**Lemma 1.** *For an L-smooth function $f(w)$ the following holds where $w^*$ is the optimal minima:*

$$f(w) - f(w^*) \geq \frac{1}{2L}||\nabla f(w)||^2$$

*Proof.* The L-smooth function by definition satisfies the following:

$$f(w^*) \leq f(v) \leq f(w) + \nabla f(w)(v - w) + \frac{L}{2}||v - w||^2$$

Now we minimize the upper bound wrt $v$ to get a tight bound on $f(w^*)$.

$$D(v) = f(w) + \nabla f(w)(v - w) + \frac{L}{2}||v - w||^2$$

after doing $\nabla_v D(v) = 0$ we get:

$$v = w - \frac{1}{L}\nabla f(w)$$

By substituting the value of $v$ in the upper bound we get:

$$f(w^*) \leq f(w) - \frac{1}{2L}||\nabla f(w)||^2$$

Hence rearranging the above term gives the desired result:

$$f(w) - f(w^*) \geq \frac{1}{2L}||\nabla f(w)||^2$$

$\square$

**Theorem 2.** *For a given classifier $h_\theta$ and one step of (steepest) gradient ascent i.e. $\Phi' = \Phi + \eta(\nabla d_{S,T}^\Phi / ||\nabla d_{S,T}^\Phi||)$ and $\Phi'' = \Phi + \eta(\nabla d_{S,T}^\Phi|_{\Phi + \hat{\epsilon}(\Phi)} / ||\nabla d_{S,T}^\Phi|_{\Phi + \hat{\epsilon}(\Phi)}||)$ for maximizing*

$$d_{S,T}^{\Phi'} - d_{S,T}^{\Phi''} \leq \eta(1 - \cos\alpha)\sqrt{2L(d_{S,T}^* - d_{S,T}^\Phi)} \tag{S4}$$

*where $\alpha$ is the angle between $\nabla d_{S,T}^\Phi$ and $\nabla d_{S,T}^\Phi|_{\Phi + \hat{\epsilon}(\Phi)}$.*

*Proof of Theorem 2.* We assume that the function is $L$-smooth (the assumption of L-smoothness is the basis of many results in non-convex optimization (Carmon et al., 2020)) in terms of input $x$. As for a fixed $h_\theta$ as we use a reverse gradient procedure for measuring the discrepancy, only one step analysis is shown. This is because only a single step of gradient is used for estimating discrepancy $d_{S,T}^\Phi$ i.e. one step of each min and max optimization is performed alternatively for optimization. After this the $h_\theta$ is updated to decrease the discrepancy. Any differential function can be approximated by the linear approximation in case of small $\eta$:

$$d_{S,T}^{\Phi + \eta v} \approx d_{S,T}^\Phi + \eta \nabla d_{S,T}^{\Phi T} v \tag{S5}$$

The dot product between two vectors can be written as the following function of norms and angle $\theta$ between those:

$$\nabla d_{S,T}^{\Phi T} v = ||\nabla d_{S,T}^\Phi|| \, ||v|| \, \cos\theta \tag{S6}$$

The steepest value will be achieved when $\cos\theta = 1$ which is actually $v = \frac{\nabla d_{S,T}^\Phi(x)}{||\nabla d_{S,T}^\Phi(x)||}$. Now we compare the descent in another direction $v_2 = \frac{\nabla d_{S,T}^\Phi|_{w+\epsilon(w)}}{||\nabla d_{S,T}^\Phi|_{w+\epsilon(w)}||}$ from the gradient descent. The difference in value can be characterized by:

$$d_{S,T}^{\Phi + \eta v} - d_{S,T}^{\Phi + \eta v_2} = \eta ||\nabla d_{S,T}^\Phi||(1 - \cos\alpha) \tag{S7}$$

As $\alpha$ is an angle between $\nabla d_{S,T}^\Phi|_{w+\epsilon(w)}$ $(v_2)$ and $\nabla d_{S,T}^\Phi(X)$ $(v)$. The suboptimality is dependent on the gradient magnitude. We use the following result to show that when optimality gap $d_{S,T}^* - d_{S,T}^\Phi(x)$ is large the difference between two directions is also large.

For an L-smooth function the following holds according to Lemma 1:

$$f(w) - f(w^*) \geq \frac{1}{2L}||\nabla f(w)||^2$$

As we are performing gradient ascent $f(w) = -d_{s,t}^{\Phi}$, we get the following result:

$$(d_{S,T}^* - d_{S,T}^{\Phi}) \geq \frac{1}{2L}||\nabla d_{S,T}^{\Phi}(x)||^2$$

$$2L(d_{S,T}^* - d_{S,T}^{\Phi}) \geq \frac{(d_{S,T}^{\Phi+\eta v_2} - d_{S,T}^{\Phi+\eta v})^2}{(\eta(1-\cos\alpha))^2}$$

$$\eta(1-\cos\alpha)\sqrt{2L(d_{S,T}^* - d_{S,T}^{\Phi})} \geq (d_{S,T}^{\Phi'} - d_{S,T}^{\Phi''})$$

This shows that difference in value of by taking a step in direction of gradient $v$ vs taking the step in a different direction $v_2$ is upper bounded by the $d_{S,T}^* - d_{S,T}^{\Phi}(x)$, hence if we are far from minima the difference can be potentially large. As we are only doing one step of gradient ascent $d_{S,T}^* - d_{S,T}^{\Phi}$ will be potentially large, hence can lead to suboptimal measure of discrepancy. □

**Theorem 3.** *Suppose $l$ is the loss function, we denote $\lambda^* := R_S^l(h^*) + R_T^l(h^*)$ and let $h^*$ be the ideal joint hypothesis:*

$$R_T^l(h_\theta) \leq \max_{||\epsilon|| \leq \rho} \hat{R}_S^l(h_{\theta+\epsilon}) + D_{h_\theta,H}^{\phi}(P_S||P_T) + \gamma(||\theta||_2^2/\rho^2) + \lambda^*. \tag{S8}$$

*where $\gamma : \mathbb{R}^+ \to \mathbb{R}^+$ is a strictly increasing function.*

*Proof of Theorem 3:* In this case we make use of Theorem 2 in the paper sharpness aware minimization (Foret et al., 2021) which states the following: The source risk $R_S(h)$ is bounded using the following PAC-Bayes generalization bound for any $\rho$ with probability $1 - \delta$:

$$R_S(h_\theta) \leq \max_{||\epsilon|| \leq \rho} \hat{R}_S(h_\theta) + \sqrt{\frac{k \log\left(1 + \frac{||\boldsymbol{\theta}||_2^2}{\rho^2}\left(1 + \sqrt{\frac{\log(n)}{k}}\right)^2\right) + 4\log\frac{n}{\delta} + \tilde{O}(1)}{n-1}} \tag{S9}$$

here $n$ is the training set size used for calculation of empirical risk $\hat{R}_S(h)$, $k$ is the number of parameters and $||\theta||_2$ is the norm of the weight parameters. The second term in equation can be abbreviated as $\gamma(||\theta||_2)$. Hence,

$$R_S(h_\theta) \leq \max_{||\epsilon|| \leq \rho} \hat{R}_S(h_\theta) + \gamma(||\theta||_2^2/\rho^2) \tag{S10}$$

From the generalization bound for domain adaptation for any f-divergence (Acuna et al., 2021) (Theorem 2) we have the following result.

$$R_T^l(h_\theta) \leq R_S^l(h_\theta) + \mathcal{D}_{h_\theta,H}^{\phi}(P_S||P_T) + \lambda^* \tag{S11}$$

Combining the above two inequalities gives us the required result we wanted to prove i.e.

$$R_T^l(h_\theta) \leq \tilde{R}_S^l(h_\theta) + D_{h_\theta,H}^{\phi}(P_S||P_T) + \gamma(||\theta||_2^2/\rho^2) + \lambda^*. \tag{S12}$$

□

## D  HESSIAN ANALYSIS

We use the PyHessian library (Yao et al., 2020) to calculate the Hessian eigenvalues and the Hessian Eigen Spectral Density. All the calculations are performed using 50% of the source data at the last checkpoint. Only the source class loss is used for calculating to clearly illustrate our point. The partition was selected randomly, and the same partition was used across all the runs. We also made sure to use the same environment to run all the Hessian experiments. A subset of the data was used for Hessian calculation mainly because the hessian calculation is computationally expensive (Yao et al., 2020). This is commonly done in hessian experiments. For example, (Chen et al., 2021) (refer Appendix D) uses 10% of training data for Hessian Eigenvalue calculation The PyHessian library uses Lanczos algorithm (Ghorbani et al., 2019) for calculating the Eigen Spectral density of the Hessian and uses the Hutchinson method to calculate the trace of the Hessian efficiently.

Table S2: Architecture used for feature classifier and Domain classifier. $C$ is the number of classes. Both classifiers will take input from feature generator ($g_\theta$).

| Layer | Output Shape |
|---|---|
| **Feature Classifier ($f_\Theta$)** | |
| - | Bottleneck Dimension |
| Linear | $C$ |
| **Domain Classifier ($\mathcal{D}_\Phi$)** | |
| - | Bottleneck Dimension |
| Linear | 1024 |
| BatchNorm | 1024 |
| ReLU | 1024 |
| Linear | 1024 |
| BatchNorm | 1024 |
| ReLU | 1024 |
| Linear | 1 |

Table S3: Accuracy (%) on VisDA-2017 (ResNet-101).

| Method | Synthetic $\rightarrow$ Real |
|---|---|
| DANN (Ganin et al., 2016) | 57.4 |
| MCD (Saito et al., 2018b) | 71.4 |
| CDAN (Long et al., 2018) | 73.7 |
| CDAN*[a] | 76.6 |
| CDAN w/ SDAT | 78.3 |
| CDAN+MCC (Jin et al., 2020) | 80.4 |
| CDAN+MCC w/ SDAT | **81.2** |

[a]Our implementation of CDAN. Refer to Section F for more details.

## E    SMOOTHNESS OF DISCRIMINATOR IN SNGAN

We also did the similar experiment of smoothing discriminator in DAT (Sec. 4.1) for SNGAN Miyato et al. (2018) as the adversarial objective in GAN is similar to DAT. We use the same configuration for SNGAN as described in PyTorchStudioGAN (Kang & Park, 2020) for both CIFAR10 (Krizhevsky et al., 2009) and TinyImageNet [3] with batch size of 256 in both cases. We then smooth the discriminator while discriminator is trained by using the same formulation as in Eq. 9. We find that smoothing discriminator leads to higher (suboptimal) Fréchet Inception Distance in case of GANs as well, shown in Fig. 3.

## F    EXPERIMENTAL DETAILS

### F.1    IMAGE CLASSIFICATION

**Office-Home**: For CDAN methods, we train the models using mini-batch stochastic gradient descent (SGD) with a batch size of 32 and a learning rate of 0.01. The learning rate schedule is the same as (Ganin et al., 2016). We train it for a total of 30 epochs with 1000 iterations per epoch. The momentum parameter in SGD is set to 0.9 and a weight decay of 0.001 is used. For CDAN+MCC experiments, we use a temperature parameter (Jin et al., 2020) of 2.5. The bottleneck dimension for the features is set to 2048.
**VisDA-2017**: We use a ResNet-101 backbone initialized with ImageNet weights for VisDA-2017 experiments. Center Crop is also used as an augmentation during training. We use a bottleneck dimension of 256 for both algorithms.
For CDAN runs, we train the model for 30 epochs with same optimizer setting as that of Office-Home. For CDAN+MCC runs, we use a temperature parameter of 3.0 and a learning rate of 0.002.
**DomainNet**: We use a ResNet-101 backbone initialized with ImageNet weights for DomainNet experiments. We run all the experiments for 30 epochs with 2500 iterations per epoch. The other parameters are the same as that of Office-Home.

To show the effectiveness of SDAT fairly and promote reproducibility, we run with and without SDAT on the same GPU and environment and with the same seed. All the above experiments were run on Nvidia V100 and RTX 2080 GPUs. We used Wandb (Biewald, 2020) to track our experiments. We will be releasing the code to promote reproducible research.

### F.1.1    ARCHITECTURE OF DOMAIN DISCRIMINATOR

One of the major reasons for increased accuracy in Office-Home baseline CDAN compared to reported numbers in the paper is the architecture of domain classifier. The main difference is the use

---

[3]https://www.kaggle.com/c/tiny-imagenet

Table S5: Accuracy(%) on **DomainNet** dataset for unsupervised domain adaptation (ResNet-101) across five distinct domains. The row indicates the source domain and the columns indicate the target domain.

| ADDA | clp | inf | pnt | rel | skt | Avg | MCD | clp | inf | pnt | rel | skt | Avg |
|---|---|---|---|---|---|---|---|---|---|---|---|---|---|
| clp | - | 11.2 | 24.1 | 41.9 | 30.7 | 27.0 | clp | - | 14.2 | 26.1 | 45.0 | 33.8 | 29.8 |
| inf | 19.1 | - | 16.4 | 26.9 | 14.6 | 19.2 | inf | 23.6 | - | 21.2 | 36.7 | 18.0 | 24.9 |
| pnt | 31.2 | 9.5 | - | 39.1 | 25.4 | 26.3 | pnt | 34.4 | 14.8 | - | 50.5 | 28.4 | 32.0 |
| rel | 39.5 | 14.5 | 29.1 | - | 25.7 | 27.2 | rel | 42.6 | 19.6 | 42.6 | - | 29.3 | 33.5 |
| skt | 35.3 | 8.9 | 25.2 | 37.6 | - | 26.7 | skt | 41.2 | 13.7 | 27.6 | 34.8 | - | 29.3 |
| Avg | 31.3 | 11.0 | 23.7 | 36.4 | 24.1 | 25.3 | Avg | 35.4 | 15.6 | 29.4 | 41.7 | 27.4 | 29.9 |
| **CDAN** | clp | inf | pnt | rel | skt | Avg | **CDAN w/ SDAT** | clp | inf | pnt | rel | skt | Avg |
| clp | - | 20.6 | 38.9 | 56.0 | 44.9 | 40.1 | clp | - | 22.0 | 41.5 | 57.5 | 47.2 | 42.1 |
| inf | 31.5 | - | 29.3 | 43.6 | 26.3 | 32.7 | inf | 33.9 | - | 30.3 | 48.1 | 27.9 | 35.0 |
| pnt | 44.1 | 19.8 | - | 57.2 | 39.9 | 40.2 | pnt | 47.5 | 20.7 | - | 58.0 | 41.8 | 42.0 |
| rel | 55.8 | 24.4 | 53.2 | - | 42.3 | 43.9 | rel | 56.7 | 25.1 | 53.6 | - | 43.9 | 44.8 |
| skt | 56.0 | 20.7 | 45.3 | 54.9 | - | 44.2 | skt | 58.7 | 21.8 | 48.1 | 57.1 | - | 46.4 |
| Avg | 46.9 | 21.4 | 41.7 | 52.9 | 38.3 | 40.2 | Avg | 49.2 | 22.4 | 43.4 | 55.2 | 40.2 | **42.1** |

of batch normalization layer in domain classifier, which was done in the library (Junguang Jiang & Long, 2020). Table S2 shows the architecture of the feature classifier and domain classifier.

### F.2 ADDITIONAL IMPLEMENTATIONS DETAILS FOR DA FOR OBJECT DETECTION

In SDAT, we modified the loss function present in Chen et al. (2018) by adding classification loss smoothing, i.e. smoothing classification loss of RPN and ROI, used in Faster R-CNN (Ren et al., 2015), by training with source data. Similarly, we applied smoothing to regression loss and found it to be less effective. We implemented SDAT for object detection using Detectron2 (Wu et al., 2019). We fixed $\rho$ to 0.15 for object detection experiments.

## G ADDITIONAL RESULTS

**VisDA-2017**: Table S3 shows the overall accuracy on the VisDA-2017 with ResNet-101 backbone. The accuracy reported in this table is the overall accuracy of the dataset, whereas the accuracy reported in the Table 5 of the main paper refers to the mean of the accuracy across classes. CDAN w/ SDAT outperforms CDAN by 1.7%, showing the effectiveness of SDAT in large scale Synthetic → Real shifts. With CDAN+MCC as the backbone, adding SDAT improves the performance of the method to 81.2%.

**DomainNet**: Table S5 shows the results of the proposed method on DomainNet across five domains. We compare our results with ADDA and MCD and show that CDAN achieves much higher performance on DomainNet compared to other techniques. It can be seen that CDAN w/ SDAT further improves the overall accuracy on DomainNet by 1.8%.

**Results with DANN** (Ganin & Lempitsky, 2015): Domain Adversarial neural networks introduced the concept of adversarial training in domain adaptation and is a seminal paper in the field of domain adaptation. Table S4 shows the results on some splits on Office-Home with DANN and DANN w/ SDAT. DANN w/ SDAT improves upon the performance on DANN specifically in challenging splits like Clipart → Art where DANN w/ SDAT gets a 1% increase over DANN.

We have shown results with three different domain adaptation algorithms namely DANN (Ganin & Lempitsky, 2015), CDAN (Long et al., 2018) and CDAN+MCC (Jin et al., 2020). SDAT has shown to improve the performance of all the three DA methods. This shows that SDAT is a generic method that can applied on top of any domain adversarial training based method to get better performance.

Table S4: Results on Office-Home dataset with DANN (Ganin & Lempitsky, 2015). DANN w/ SDAT improves the performance over DANN across the four splits of Office-Home dataset showing the adaptability of the proposed method.

| Method | Ar→Cl | Cl→Pr | Rw→Cl | Pr→Cl | Average |
|---|---|---|---|---|---|
| DANN (Ganin & Lempitsky, 2015) | 52.6 | 65.4 | 60.4 | 52.3 | 57.7 |
| DANN w/ SDAT | **53.4** | **66.4** | **61.3** | **53.8** | **58.7** |

## H    DIFFERENT SMOOTHING TECHNIQUES

**Stochastic Weight Averaging (SWA)** (Izmailov et al., 2018): SWA is a widely popular technique to reach a flatter minima. The idea behind SWA is that averaging weights across epochs leads to better generalization because it reaches a wider optima. The recently proposed SWA-Densely (SWAD) (Cha et al., 2021) takes this a step further and proposes to average the weights across iterations instead of epochs. SWAD shows improved performance on domain generalization tasks. We average every 400 iterations in the SWA instead of averaging per epochs. We tried averaging across 800 iterations as well and the performance was comparable.

**Virtual Adversarial Training (VAT)** (Miyato et al., 2019): VAT is regularization technique which makes use of adversarial perturbations. Adversarial perturbations are created using Algo. 1 present in (Miyato et al., 2019). We added VAT by optimizing the following objective:

$$\min_\theta \mathbb{E}_{x \sim P_S}[\max_{||r|| \leq \epsilon} D_{KL}(h_\theta(x)||h_\theta(x+r))] \tag{S13}$$

This value acts as a negative measure of smoothness and minimizing this will make the model smooth. For training, we set hyperparameters $\epsilon$ to 15.0, $\xi$ to 1e-6, and $\alpha$ as 0.1.

**Label Smoothing (LS)** (Szegedy et al., 2016): The idea behind label smoothing is to have a distribution over outputs instead of one hot vectors. Assuming that there are k classes, the correct class gets a probability of 1 - $\alpha$ and the other classes gets a probability of $\alpha$ / (k-1). (Stutz et al., 2021) mention that label smoothing tends to avoid sharper minima during training. We use a smoothing parameter ($\alpha$) of 0.1 in all the experiments in Table S6. We also show results with smoothing parameter of 0.2 and observe comparable performance. We observe that label smoothing slightly improves the performance over DAT.

Table S6: Different Smoothing techniques. We refer to (Stutz et al., 2021) to compare the proposed SDAT with other techniques to show the efficacy of SDAT. It can be seen that SDAT outperforms the other smoothing techniques significantly. Other smoothing techniques improve upon the performance of DAT showing that smoothing is indeed necessary for better adaptation.

| Method | Ar→Cl | Cl→Pr | Rw→Cl | Pr→Cl |
|---|---|---|---|---|
| DAT | 54.3 | 69.5 | 60.1 | 55.3 |
| VAT | 54.6 | 70.7 | 60.8 | 54.4 |
| SWAD-400 | 54.6 | 71.0 | 60.9 | 55.2 |
| LS ($\alpha = 0.1$) | 53.6 | 71.6 | 59.9 | 53.4 |
| LS ($\alpha = 0.2$) | 53.5 | 71.2 | 60.5 | 53.2 |
| SDAT | **55.9** | **73.2** | **61.4** | **55.9** |

## I    OPTIMUM RHO VALUE

Table S7 and S8 show that $\rho$ = 0.02 works robustly across experiments providing an increase in performance (although it does not achieve the best result each time) and can be used as a rule of thumb.

Table S7: $\rho$ value for DomainNet

| Split | DAT | SDAT($\rho = 0.02$) | SDAT - Reported ($\rho = 0.05$) |
|---|---|---|---|
| **clp→skt** | 44.9 | 46.7 | 47.2 |
| **skt→clp** | 56.0 | 59.0 | 58.7 |
| **skt→pnt** | 45.3 | 47.8 | 48.1 |
| **inf→rel** | 43.6 | 47.3 | 48.1 |

Table S8: $\rho$ value for VisDA-2017 Synthetic → Real

| Backbone | DAT | SDAT ($\rho = 0.02$) | SDAT Reported($\rho = 0.005$) |
|---|---|---|---|
| CDAN | 76.6 | 78.2 | 78.3 |
| CDAN+MCC | 80.4 | 80.9 | 81.2 |

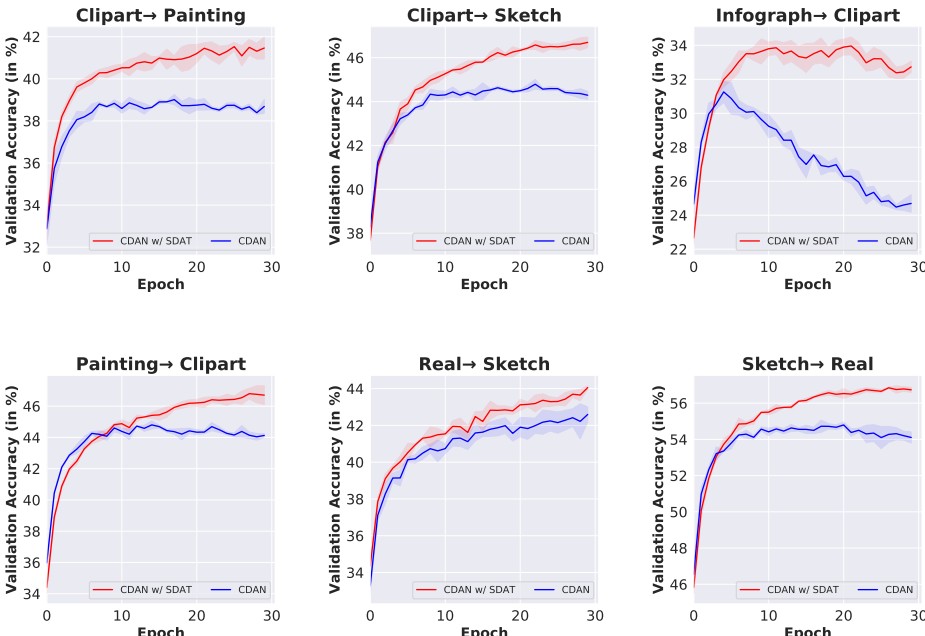

Figure S1: Validation Accuracy across epochs on different splits of DomainNet. We run on three different random seeds and plot the error bar indicating standard deviation across runs. CDAN w/ SDAT consistently outperforms CDAN across different splits of DomainNet.

## J  SIGNIFICANCE AND STABILITY OF EMPIRICAL RESULTS

To establish the empirical results' soundness and reliability, we run a subset of experiments (representative of each different source domain) on DomainNet. The experiments are repeated with three different random seeds leading to overall 36 experimental runs (18 for CDAN w/ SDAT (Our proposed method) and 18 for CDAN baseline). Due to the large computational complexity of each experiment (≈20 hrs each), we have presented results for multiple trials on a subset of splits. We find (in Table S9) that our method can outperform the baseline average in each of the 6 cases, establishing significant improvement across all splits. However, we found that due to the large size of DomainNet, the average increase (across three different trials) is close to the reported increase in all cases (Table S9), which also serves as evidence of the soundness of reported results (for remaining splits). We also present additional statistics below for establishing soundness.

If the proposed method is unstable, there is a large variance in the validation accuracy across epochs. For analyzing the stability of SDAT, we show the validation accuracy plots in Figure S1 on six different splits of DomainNet. We find that our proposed SDAT improves over baselines consistently across epochs without overlap in confidence intervals in later epochs. This also provides evidence for the authenticity and stability of our results. We also find that in some cases, like when using the Infographic domain as a source, our proposed SDAT also significantly *stabilizes the training* (Figure S1 Infographic → Clipart).

One of the other ways of reporting results reliably proposed by the concurrent work (Berthelot et al., 2021) (Section 4.4) involves reporting the median of accuracy across the last few checkpoints. The median is a measure of central tendency which ignores outlier results. We also report the median of validation accuracy for our method *across all splits* for the last five epochs. It is observed that we observe similar gains for median accuracy (in Table S10) as reported in Table 2.

Table S9: DomainNet experiments over 3 different seeds. We report the mean, standard deviation, reported increase and average increase in the accuracy (in %).

| Split | CDAN | CDAN w/ SDAT | Reported Increase (Table 2) | Average Increase |
|-------|------|--------------|-----------------------------|------------------|
| **clp→pnt** | $38.9 \pm 0.1$ | $41.5 \pm 0.3$ | +2.6 | +2.6 |
| **skt→rel** | $55.1 \pm 0.2$ | $57.1 \pm 0.1$ | +2.2 | +2.0 |
| **pnt→clp** | $44.5 \pm 0.3$ | $47.1 \pm 0.3$ | +3.4 | +2.6 |
| **rel→skt** | $42.4 \pm 0.4$ | $43.9 \pm 0.1$ | +1.6 | +1.5 |
| **clp→skt** | $44.9 \pm 0.2$ | $47.3 \pm 0.1$ | +2.3 | +2.4 |
| **inf→clp** | $31.4 \pm 0.5$ | $34.2 \pm 0.3$ | +2.3 | +2.7 |

Table S10: Median accuracy of last 5 epochs on DomainNet dataset with CDAN w/ SDAT. The number in the parenthesis indicates the increase in accuracy with respect to CDAN.

| Target (→) Source (↓) | clp | inf | pnt | real | skt | Avg |
|------------------------|-----|-----|-----|------|-----|-----|
| **clp** | - | 21.9 (+1.7) | 41.6 (+3.0) | 56.5 (+1.3) | 46.4 (+2.0) | 41.6 (+2.0) |
| **inf** | 32.4 (+7.9) | - | 29.8 (+7.0) | 46.7 (+12.7) | 25.6 (+5.4) | 33.6 (+8.2) |
| **pnt** | 47.2 (+2.9) | 21.0 (+1.1) | - | 57.6 (+1.0) | 41.5 (+2.4) | 41.8 (+1.8) |
| **real** | 56.5 (+0.7) | 25.5 (+0.9) | 53.9 (+0.5) | - | 43.5 (+1.3) | 44.8 (+0.8) |
| **skt** | 59.1 (+3.0) | 22.1 (+1.7) | 48.2 (+3.1) | 56.6 (+2.9) | - | 46.5 (+2.7) |
| **Avg** | 48.8 (+3.6) | 22.6 (+1.3) | 43.4 (+3.4) | 54.3 (+4.5) | 39.2 (+2.8) | 41.7 (+3.1) |

As the Office-Home dataset is smaller (i.e., 44 images per class) in comparison to DomainNet we find that there exists some variance in baseline CDAN results (This is also reported in the well-known benchmark for DA (Junguang Jiang & Long, 2020)). For establishing the empirical soundness, we report results of 4 different dataset splits on three different random seeds. It can be seen in Table S11 that even though there is variance in baseline results, our combination of CDAN w/ SDAT can produce consistent improvement across different random seeds. This further establishes the empirical soundness of our procedure.

Table S11: Office-Home experiments over 3 different seeds. We report the mean, standard deviation, reported increase and average increase in the accuracy (in %).

| Split | CDAN | CDAN w/ SDAT | Reported Increase (Table 1) | Average Increase |
|-------|------|--------------|-----------------------------|------------------|
| **Ar→Cl** | $53.9 \pm 0.2$ | $55.5 \pm 0.2$ | +1.7 | +1.6 |
| **Ar→Pr** | $70.6 \pm 0.4$ | $72.1 \pm 0.4$ | +1.6 | +1.5 |
| **Rw→Cl** | $60.7 \pm 0.5$ | $61.8 \pm 0.4$ | +1.3 | +1.1 |
| **Pr→Cl** | $54.7 \pm 0.4$ | $55.5 \pm 0.4$ | +0.6 | +0.8 |

