# OpenReview forum: "A Closer Look at Smoothness in Domain Adversarial Training"
_ICLR.cc/2022/Conference — ICLR 2022 Submitted_

### Official Review · Reviewer_nwci · 2021-11-01

**Correctness:** 4
**Technical Novelty And Significance:** 4
**Empirical Novelty And Significance:** 4
**Recommendation:** 8
**Confidence:** 4

**Main Review:**

STRENGTHS:
- The paper is very well-written and introduces all used concepts appropriately. This makes the paper easy to understand even for readers that are not familiar with domain adaption.
- The motivational experiments performed in Section 4 (Analysis of Smoothness) are well-explained and convincing.
- The author(s) provide(s) a generalization bound (Theorem 2) for their proposed method which justifies it from a theoretical perspective.
- The experimental evaluation is very extensive: the method is evaluated on three different domain adaption datasets for classification (Office-Home, VisDA-2017, and DomainNet) and compared with different existing domain adaption methods. It is shown that SDAT can be combined with the existing methods CDAN and CDAN + MCC to increase generalization performance.
- Furthermore, improved performance on domain adaption datasets for object detection is reported.
- Additional experiments show that SDAT also increases the robustness to label noise and that SAM outperforms other smoothing techniques.

WEAKNESSES:
- I see not major issues with this submission.

MINOR REMAKS:
- Some references are not correctly capitalized (e.g., "Lower bounds for finding stationary points i" or "Faster r-cnn") also conference names are sometimes capitalized and sometimes not.

**Summary Of The Paper:**

The paper analyzes the role of smoothness in domain adversarial training (DAT). The main insight of the paper is that smoother minima of the classification loss improve generalization in the target domain. This explains why SGD is usually preferred over Adam when optimizing the training objective of DAT. To further improve the generalization performance by leveraging this phenomenon the author(s) introduce(s) smooth domain adversarial training (SDAT) for classification and object detection. SDAT applies sharpness aware minimization (SAM) to find smoother minimia of the classification loss used during DAT.

**Summary Of The Review:**

Overall, I would highly recommend to accept this submission to ICLR 2022. The theoretical and empirical results are convincing and the paper is of high quality.

---

> ### Author Response · Authors · 2021-11-15
> **Response to Reviewer nwci**
>
> We thank the reviewer for the detailed feedback and encouraging comments on our paper. We are happy to see that all aspects which we put effort into, were recognized. We will fix all the minor changes suggested by you in the revised version.

---

### Official Review · Reviewer_C1yh · 2021-11-02

**Correctness:** 4
**Technical Novelty And Significance:** 2
**Empirical Novelty And Significance:** 2
**Recommendation:** 5
**Confidence:** 2

**Main Review:**

Strengths:

  * Theoretical foundation for smoothness
  * Results sections include improvements on both image classification and object detection.
  * Ablation experiments provide insights to the effect of a smoothness parameter.

Weaknesses:

  * Figure 4: not clear why and how the optimum for rho differs per dataset. Does this depend on the dataset size or diversity of objects? Moreover, the vanilla ResNets have a batchnorm layer, so it is not clear why figure 4a has such a sharp optimum at all.
* What is the final recommendation of the paper? If the paper recommends using smoothness penalties, then what are the limits? For example, how does the smoothness of the classifier (focus of this study) interplay with smoothness of the domain discriminator? I could imagine that when the discriminator has sharp decision boundaries, then enforcing smoothness on the classifier would be less useful. (But in practise people employ many tricks to get a smooth discriminator [1][2]).
* Table 1: most improvements occur when combining SDAT with MCC. This combination is not well explained in the paper. What makes these approaches complementary and why are improvements small (or non-existent) for vanilla CDAN?
* Theorem 2: I don’t see how this theorem relates to the point of the paper. From equation 11 it seems smoothness is applied to h_\theta.

[1] Gulrajani et al. "Improved training of wasserstein gans." arXiv 2017.

[2] Arjovsky et al. "Towards principled methods for training generative adversarial networks." arXiv 2017.

[3] Miyato et al. "Spectral normalization for generative adversarial networks." arXiv 2018.


**Summary Of The Paper:**

This paper introduces a smoothness penalty for domain adversarial training. The penalty encourages smoothness on the parameter space of a discriminative model. This smoothness penalty is motivated by results in Sharpness Aware Minimization. Results are included on the popular DA datasets such as Office-home and VisDA, where superior results are shown. Ablation experiments show how different smoothness penalties change the observed improvements.

**Summary Of The Review:**

Paper with theoretical motivation and results on both classification and object detection. However, the limitations of the proposed method are not clear. When would smoothness not be useful and how do the improvements depend on the other tricks for training Domain Adaptation models.

---

> ### Author Response · Authors · 2021-11-15
> **Response to Reviewer C1yh [1/2]**
>
> We thank the reviewer for the detailed comments and the interesting questions.
> > Figure 4: not clear why and how the optimum for rho differs per dataset. Does this depend on the dataset size or diversity of objects? Moreover, the vanilla ResNets have a batchnorm layer, so it is not clear why figure 4a has such a sharp optimum at all.
>
> The different datasets contain varying degrees of domain shift, i.e. (VisDA adapts from synthetic domain to real-world data, whereas Office adapts from (Art domain to Product domain)). Hence, the image pattern differs a lot from dataset to dataset, which is why different $\rho$ values exist. We want to clarify that the performance difference between $\rho$ = 0.02 and $\rho$=0.05 is low in Figure 4a, but due to the scale of the graph used, it appears that a sharp minimum is present. We would also like to state that the value of $\rho$ = 0.02 works robustly across experiments providing an increase in performance (although it does not achieve the best result each time) and can be used as a rule of thumb. Thanks for pointing out the scale, we will update it in the next version. We have also added these tables in the Appendix I (Table S7 and S8) of the paper.
>
> $\rho$ value ablation for DomainNet with CDAN backbone.
>
> | Split      | DAT | SDAT ($\rho$ = 0.02) | SDAT - Reported($\rho$ = 0.05) |
> | :---        |:----: |    :----:   |          ---: |
> | c -> s     | 44.9   | 46.7      |  47.2      |
> | s -> c     | 56.0   | 59.0      |  58.7      |
> | s -> p     | 45.3   | 47.8      |  48.1      |
> | i -> r     | 43.6   | 47.3      |  48.1      |
>
> $\rho$ value for VisDA-2017 dataset.
>
> | Backbone      | DAT | SDAT ($\rho$ = 0.02) | SDAT - Reported($\rho$ = 0.005) |
> | :---        |:----: |    :----:   |          ---: |
> | CDAN     | 76.6   | 78.2      |  78.3      |
> | CDAN+MCC    | 80.4   | 80.9      |  81.2      |
>
> > What is the final recommendation of the paper? If the paper recommends using smoothness penalties, then what are the limits? For example, how does the smoothness of the classifier (focus of this study) interplay with smoothness of the domain discriminator? I could imagine that when the discriminator has sharp decision boundaries, then enforcing smoothness on the classifier would be less useful. (But in practise people employ many tricks to get a smooth discriminator [1][2]).
>
> The final recommendation of the paper is that smoothing the discriminator leads to sub-optimal performance. More generally, in the case of multi-objective optimization, smoothing all the terms might not necessarily lead to better performance. We find that the configuration when only classifier converges to smooth optima and discriminator is not smooth produces the best results. We have provided a comparison in Table 4C of the updated draft, as well as in the response to Reviewer zPwA. With respect to the limit, we refer the reviewer to Figure 4A, where we show the ablation on $\rho$. We observed that as the smoothness increases ($\rho$ value), the performance improves to a certain point. Further increasing the smoothness would degrade the performance.
>
> Thanks for the interesting question on GANs. From earlier work [1][2], one would expect that a smooth discriminator using SAM would be useful. But we find that the opposite of that expectation is true, which makes our result **interesting**. However, we believe this difference comes from the fact that SAM regularizes the trajectory of SGD, whereas other penalties regularize the model weights **W**. We find that SAM regularizing trajectories for converging to smooth optima slows down GAN training (Figure 3c), hence converging to a suboptimal GAN. Hence our conclusion that smoothing discriminator using techniques like SAM can be detrimental also holds for GANs, establishing the generality of our result.
>
> > Table 1: most improvements occur when combining SDAT with MCC. This combination is not well explained in the paper. What makes these approaches complementary and why are improvements small (or non-existent) for vanilla CDAN?
>
> We would like to kindly clarify that the improvements of SDAT are comparable for both the cases, i.e. (CDAN w/ SDAT achieves 69.5% whereas vanilla CDAN achieves 68.4% average accuracy) which translates to 1.1% improvement, which is comparable to a 0.9% improvement on average on CDAN+MCC in Table 1. We would also like to add that in the case of VisDA-2017, vanilla CDAN + SDAT (Our method) leads to an effective improvement of 1.5% (Table 2). In all cases, we found that the improvement due to SDAT over vanilla CDAN **is either similar or better** than improvement over CDAN+MCC.

---

> > ### Author Response · Authors · 2021-11-15
> > **Response to Reviewer C1yh [2/2]**
> >
> > > Theorem 2: I don't see how this theorem relates to the point of the paper. From equation 11 it seems smoothness is applied to h_\theta.
> >
> > As there are two components in the loss i.e. classification (or task loss) and the adversarial loss, naively applying smoothing to both doesn't lead to best results. Theorem 2 infers that smoothing $d^{\Phi}\_{s,t}$ leads to suboptimal results (empirically also shown in Figure 3)). Hence, that's why we only smooth $h\_{\theta}$, which is the objective that leads to the best results in Equation 11.
> >
> >
> > > When would smoothness not be useful and how do the improvements depend on the other tricks for training Domain Adaptation models.
> >
> > From Theorem 2, we have shown that smoothing adversarial components would lead to suboptimal performance, hence smoothness of SAM won't be helpful in such cases. This is also observed in the empirical results observed in Figure 3C (GANs) and results on domain adaptation (Fig. 3B).
> >
> > Our improvements are based on converging to a smoother optimum. This is why our method can easily be combined with other methods (like CDAN+MCC) to improve performance. To show the generality of the improvements, we have run the experiments with both CDAN and CDAN+MCC, wherein **both cases we achieve significant improvement**. Our method is effective on both vanilla versions of CDAN and CDAN+MCC as well, and the improvement is not dependent on MCC (or any other tricks).
> >
> > We hope that our clarifications would clear out that our method can is equally effective over vanilla CDAN and doesn't require any other additional tricks for improvement. Please feel free to get back to us in case you have further questions.

---

### Official Review · Reviewer_zPwA · 2021-11-02

**Correctness:** 3
**Technical Novelty And Significance:** 2
**Empirical Novelty And Significance:** 2
**Recommendation:** 5
**Confidence:** 3

**Main Review:**

## Strong points

1. The paper takes issue with an interesting point of the reason for SGD optimizer outperforming Adam on DA tasks for reaching a smoother minima, develops it theoretically, and proposed a novel loss function focusing on smoothing classifier loss to improve generalization on target domain.

2. It fully discussed why we should not apply the smoothing method **on the discrepancy term**, and verify the theory empirically through experiment, further developing smooth theory with regard to **adversarial objectives.**

## Weak points

1. The method for smoothing ERM is more directly adopted from SAM rather than a novel idea, and the relation **between** smoothness **and** better generalization has also been discussed by previous papers (SWAD), therefore renders this paper not insightful enough.

2. The proposed smoothing method w.r.t discrepancy term tend to "lead to a suboptimal solution", therefore not significantly helpful when tackling DA tasks (and I think it is hard to interpret). What actually makes such difference on the finally result is still, smoothness on ERM term.

**Summary Of The Paper:**

The paper discussed a correlation of smoothness on **classification loss** and **generalization ability on target domain** and further develop a method to enhance such smoothness to achieve better performance on DA tasks. To do so, it adopted the losses based on Sharpness Aware Minimization, a minmax game of finding a smoother neighborhood of $\theta$ , by computing first order deviation. Empirical studies are implemented to verify the theory and show the soundness of the proposed method.

**Summary Of The Review:**

Overall, the author has well established a positive correlation between smoothness of classification loss and generalization capability on target domain, and a negative correlation between adversarial counterparts. The method of acquiring a flat minima also brings promising results on multiple DA classification and detection datasets. However, as I can see the experimental studies of the proposed smoothing method are well-designed and theories are rigorously proved, I am not completely convinced by the novelty of the work. Hope the author can make further analysis and explanation.

---

> ### Author Response · Authors · 2021-11-14
> **Response to Reviewer zPwA**
>
> We thank the reviewer for their detailed review and insightful comments on our paper.\
> Please find below our responses to each of the concerns raised.
>
> > The method for smoothing ERM is more directly adopted from SAM rather than a novel idea,
> and the relation between smoothness and better generalization has also been discussed by previous papers (SWAD),
> therefore renders this paper not insightful enough.
>
> Yes, we do agree that we use SAM for smoothing the ERM term. However, we would like to clarify that for tasks like Domain Adaptation, GANs, etc., the overall loss is a **combination of minimization (ERM) and Adversarial Losses**. Our paper demonstrates that *smoothing ERM terms are beneficial, whereas smoothing adversarial terms degrades performance.* Hence, our work provides a directive to users on which terms in loss should be smooth for optimal performance.
>
> **Comparison with SWAD**:
> - SWAD is focused on Domain Generalization and applied for methods that have minimization (ERM) terms. In contrast, our work is focused on Adversarial Domain Adaptation, which has the presence of both the minimization (ERM) and the adversarial terms in the loss.
> SWAD obtains smoothness by using Weight Averaging, which is orthogonal to Sharpness Aware methods which are focused on optimizing a smooth objective (hence both converge to different solutions). Hence we find it is important to test if smooth objective-based methods also converge to solutions that are helpful for Domain Adaptation.
> - As SWAD performs Weight Averaging, it is **not possible to selectively smooth only minimization (ERM) components with SWAD**, as gradients for both the adversarial loss and ERM update weights of the backbone. Due to this, SWAD cannot reach optimal performance for DAT. For verifying this, we also compare our method by implementing SWAD for Domain Adaptation on four different dataset splits in Table 5. On average, SDAT (Ours) gets 61.6% (**+2.4%** over DAT) accuracy in comparison to 60.4% (**+1.2%** over DAT) for SWAD. SWAD was also recently published at NeurIPS 2021 and was concurrent with the development of this work.
> - The theoretical results presented in both papers are orthogonal and of independent importance.
>
> > The proposed smoothing method w.r.t discrepancy term tends to "lead to a suboptimal solution",
> therefore not significantly helpful when tackling DA tasks (and I think it is hard to interpret).
> What actually makes such a difference on the final result is still, smoothness on ERM term.
>
> Yes, smoothness on the ERM term does lead to improvement in performance. However, if a smooth objective is applied to both the adversarial (discrepancy) term and ERM term (apply SAM to all terms), this does not lead to optimal results. We have provided the comparison for this in Figure 4C (SDAT w/ all refers to applying SAM on both ERM and adversarial terms). For improving clarity, we will add the table in the following format:
>
> For Ar->Cl split of the Office Home Dataset:
>
> | Smooth Classification Loss    | Smooth Adversarial Loss | Accuracy    |
> | :---      |   :----:   |          ---: |
> | ❌     | ❌         |  54.3     |
> | ❌   | ✅       | 51.0      |
> | ✅   | ❌       | **55.7**  |
> | ✅   | ✅       | 54.9      |
>
> We also performed a similar experiment on DomainNet (P2C split) where the smoothing classifier only (SDAT) obtains 47.48 in comparison to 45.19 (smoothing both classification and Adversarial Loss). Hence the fact that *only smoothing classification loss is optimal*, is important and significant for best performance.
>
>
>
> > I am not completely convinced by the novelty of the work.
>
>
> We would like to the point that in **SWAD cannot selectively smooth specific components of the loss** in case there are both adversarial and ERM terms in loss, hence won't be able to achieve optimal results. Whereas in SDAT, we are able to selectively smoothness through our optimization objective.
>
> Also, we shed light on the role of Adam vs. SGD in DAT. This is the first such work analyzing the Adam vs. SGD dilemma in DAT (where no reason has been provided for preference of SGD).
>
> We will add the difference discussion on SWAD (in the final version) and have added the above table in the updated version of the manuscript. We hope our response clarified your concerns. Please feel free to ask for any further clarifications.

---

### Official Review · Reviewer_rxW2 · 2021-11-02

**Correctness:** 3
**Technical Novelty And Significance:** 2
**Empirical Novelty And Significance:** 3
**Recommendation:** 5
**Confidence:** 4

**Main Review:**

The main contribution of this work is to propose applying Sharpness Aware Minimization (SAM) to smooth only the task loss term of domain adversarial training. From an empirical standpoint the paper is interesting and points to potentially relevant investigation directions, however, I have concerns regarding the significance of this study since there is no clear motivation as to why previous findings on the iid setting showing that smoother minima generalize better would transfer to the unsupervised domain adaptation setting. Moreover, the empirical validation of the proposed approach is not extensive enough to provide sufficient evidence for the soundness of the reported findings. In the following, I provide more details about my concerns along with questions and suggestions.

- Motivation:
  - Although the authors (to some extent) empirically show that SDAT can improve the performance on the target domain on the considered cases, there is no justification on why one would expect better out-of-distribution generalization for smoother minimum. Would the authors clarify that? More specifically, what is the motivation/context for investigating Conjecture 1? As of now, it seems to me that this work is extending an analysis for the iid setting to unsupervised domain adaptation without a clear reason.
  - It is not clear to me why the authors decided to provide results that rely on Acuna et al. 2021 rather than consider the classic results from Ben-David et al. 2010. There is no clear motivation for this choice in the manuscript and I believe the authors should better justify this choice.


- Results:
  - The relevance of both theoretical results are not clear to me. Theorem 2, for example, requires unrealistic assumptions such as L-smoothness. Theorem 3, in turn, states a bound that is looser than the known results in the literature and which are already vacuous, so it is not clear to me what is the relevance of this result in the context of unsupervised domain adaptation.
  - It is hard to assess the relevance of the reported results as no measures of statistical significance were reported. More specifically, the authors should include in the paper information regarding the number of runs considered for each experiment and report at least the average and standard deviation for all the quantities reported. Otherwise, I don’t think it is possible to rely on the current experimental set-up to support the claims of this contribution.
  - Please clarify how the 50% partition of the source data was selected to estimate the Hessian for the three compared methods and if the same partition was used for all, otherwise results on Figure 3 could be noisy.
  - SDAT requires an extra gradient step in order to estimate the smoothing penalty in Eq. 7. How does this affect the computational cost of SDAT and how SDAT stands in comparison with the considered baselines?


- Other concerns/questions:
  - Some statements of the manuscript are not rigorous and require rephrasing:
     - Abstract: “combination of classification and adversarial terms”. In the considered setting, the adversarial term of the loss also stems from a classification task (domain discrimination), therefore, it is confusing to the denominate both component losses in such a way. I suggest referring to the loss term related to classifying labels regarding a specific task as task loss.
     - Section 3.1: please specify what “works well on the dataset” means (i.e. low risk on the target distribution).
     - Theorems 1/3: please precisely define what is the ideal classifier $h^*$.
     - Section 4: “where we find that in contrast to supervised learning” notice that, in practice, unsupervised domain adaptation also corresponds to supervised learning since all losses require either task or domain labels, so it shouldn’t be referred to as a setting opposed to supervised learning.
  - Table 1 caption: what exactly “sophisticated” means here?



- Related work:
  - Foundational literature on domain adaptation is missing in the related work section, for example, Ben-David et al. (2007) and Long et al. (2016).

Ben-David et al., Analysis of representations for domain adaptation, 2007.
Long et al., Unsupervised domain adaptation with residual transfer networks, 2016.


**Summary Of The Paper:**

This work studies the loss landscape of domain adversarial neural networks for domain adaptation. The authors claim that similarly to the iid setting, smooth minima might yield better generalization on the unsupervised domain adaptation framework via domain adversarial training but in case smoothness is enforced only with respect to the task loss. Experiments on the OfficeHome dataset indicated this hypothesis might be true. The authors then proposed to apply a previously proposed approach to enforce smoothness (Sharpness Aware Minimization), so that only smoothness with respect to the task loss would be enforced. Moreover, a theorem showing that the non-smooth domain discrepancy estimation is better than the smoother counterpart was introduced, along with a generalization bound for the risk on the target domain in terms of the smooth risk on the source domain. Finally, the authors showed that the proposed approach, SDAT, improved the performance on target domains of three domain adaptation datasets in comparison to the considered baselines. Further experiments showed that SDAT performed better than other smoothing techniques, as well as yields improved robustness to label noise in comparison to not enforcing smooth minima.

**Summary Of The Review:**

This work studies the loss landscape of adversarial domain adaptation in terms of the smoothness of the minima. Despite showing somewhat empirically promising results, I found this work lacks a solid motivation for the presented analysis (i.e. why should we expect that strategies that improve generalization in the iid case would help in a non-iid setting as well?) and a sound empirical validation of the proposed approach. Moreover, the relevance of the presented generalization bound is also unclear to me at this point. All in all, I think this contribution seems promising but due to the concerns I raised in my review, I believe it is not yet ready for publication.

---

> ### Author Response · Authors · 2021-11-15
> **Response to Reviewer rxW2 [1/3]**
>
> We thank you for your effort in providing a detailed review of our work.
>
> > Although the authors (to some extent) empirically show that SDAT can improve the performance on the target domain on the considered cases, there is no justification on why one would expect better out-of-distribution generalization for smoother minimum. Would the authors clarify that? More specifically, what is the motivation/context for investigating Conjecture 1? As of now, it seems to me that this work is extending an analysis for the iid setting to unsupervised domain adaptation without a clear reason.
>
> Converging to smooth minima is shown to have generalize models robust to corruptions (which are out of distribution) in comparison to SGD [1]. [1] shows significant improvement in performance on both ImageNet-C and ImageNet-R (both out of distribution datasets). Also, smoothness in image space has been shown to be beneficial for domain adaptation [2]. We aim to explore the smoothness in weight space and its effect on domain adaptation in this current work.
>
> We have improved the motivation section based on this in Section 2 (Related Work) of the updated draft.
>
> [1] Chen, X., Hsieh, C. J., & Gong, B. (2021). When Vision Transformers Outperform ResNets without Pretraining or Strong Data Augmentations. arXiv preprint arXiv:2106.01548.\
> [2] Shu, R., Bui, H. H., Narui, H., & Ermon, S. (2018). A dirt-t approach to unsupervised domain adaptation. arXiv preprint arXiv:1802.08735.
>
> > It is not clear to me why the authors decided to provide results that rely on Acuna et al. 2021 rather than consider the classic results from Ben-David et al. 2010. There is no clear motivation for this choice in the manuscript and I believe the authors should better justify this choice.
>
> Acuna et al. 2021 (ICML) generalize the results of Ben David et al. 2010 to minimize any f-divergence between source and target distribution. All the classic results
> from Ben-David et al. 2010 can be derived from the framework of Acuna et al. 2021 as its more generic, hence we have also used it in our work.
>
>
> > The relevance of both theoretical results are not clear to me. Theorem 2, for example, requires unrealistic assumptions such as L-smoothness. Theorem 3, in turn, states a bound that is looser than the known results in the literature and which are already vacuous, so it is not clear to me what is the relevance of this result in the context of unsupervised domain adaptation.
>
> L-Smoothness assumption requires that the gradient norms are bounded with (L), which is usually true while we train our networks (gradients do not explode). L-smoothness is a fundamental assumption used for proving the convergence of non-convex objectives (e.g. neural networks) and is commonly used in works [3,4,5]. We have described the reasoning for this assumption in Appendix C (Proof 2), which we will include reference to in the main draft.
>
> We want to clarify that by Theorem 3, we do not aim to get a better generalization bound of complexity. We aimed to establish the fact that our procedure is consistent (i.e when n (number of samples) is large, the target risk will be bounded by source risk and discrepancy term). Hence in case, n is large, the target risk will get reduced through SDAT if we reduce the smooth objective and the discrepancy. This establishes that our proposed method SDAT is sound and consistent for large n, similarly as DAT. Proving generalization boun for a new DA objective is a common practice followed by recent works for demonstrating soundness [6, 10].
>
> [3] Carmon, Y., Duchi, J. C., Hinder, O., & Sidford, A. (2019). Lower bounds for finding stationary points I. Mathematical Programming, 1-50.\
> [4] Carmon, Y., Duchi, J. C., Hinder, O., & Sidford, A. (2017, July). “Convex Until Proven Guilty”: Dimension-Free Acceleration of Gradient Descent on Non-Convex Functions. In International Conference on Machine Learning (pp. 654-663). PMLR.\
> [5] Zhang, J., He, T., Sra, S., & Jadbabaie, A. (2019). Why gradient clipping accelerates training: A theoretical justification for adaptivity. arXiv preprint arXiv:1905.11881.\
> [6] Zhang, Y., Liu, T., Long, M., & Jordan, M. (2019, May). Bridging theory and algorithm for domain adaptation. In International Conference on Machine Learning (pp. 7404-7413). PMLR.\
> [10] Peng, X., Huang, Z., Zhu, Y., & Saenko, K. (2019). Federated adversarial domain adaptation. arXiv preprint arXiv:1911.02054.

---

> > ### Author Response · Authors · 2021-11-15
> > **Response to Reviewer rxW2 [2/3]**
> >
> > > It is hard to assess the relevance of the reported results as no measures of statistical significance were reported. More specifically, the authors should include in the paper information regarding the number of runs considered for each experiment and report at least the average and standard deviation for all the quantities reported. Otherwise, I don’t think it is possible to rely on the current experimental set-up to support the claims of this contribution.
> >
> > We report results on some splits of the DomainNet dataset over three different trials in the below table.
> > The results clearly show a trivial difference in performance across the three runs, indicating the proposed method's superiority and the soundness of the empirical results.\
> > We hope that this clarifies the issue of reliability of the empirical results. We plan to include this for more experiments in the final version.
> > We would also like to mention that we have run **DAT and SDAT on the same GPU with the same environment and random seeds for sound comparison** in each experiment.\
> > Furthermore, we would like to state that domain adaptation datasets usually contain multiple splits that lead to multiple pairs of experiments (eg. 20 for DomainNet, 12 for Office-Home), getting an increase due to SDAT(our method) in all these splits in itself is a sanity check for soundness. Running each experiment three times is often computationally infeasible in such cases. Hence the accuracy is reported on a single trial.(this practice is followed by all our baselines, e.g., f-DAL, SRDC, etc., for Office-Home and DomainNet datasets). We have added this table in Appendix Section J (Table S9).
> >
> >
> > | Split      | Dataset | CDAN| CDAN w/ SDAT |
> > | :---        |:----: |    :----:   |          ---: |
> > | c -> p     | DomainNet   | 38.9  ± 0.1|  41.5 ± 0.3      |
> > | s -> r     | DomainNet   | 55.1 ± 0.2      |  57.1 ±0.1      |
> > | p -> c     | DomainNet   | 44.5 ± 0.3      |  47.1 ± 0.3      |
> >
> >
> >
> > > Please clarify how the 50% partition of the source data was selected to estimate the Hessian for the three compared methods and if the same partition was used for all, otherwise results on Figure 3 could be noisy.
> >
> > Thank you for this question. The partition was selected randomly, and the same partition was used across all the runs. We also made sure to use the same environment to run all the Hessian experiments. A subset of the data was used for Hessian calculation mainly because the hessian calculation is computationally expensive [9]. This is commonly done in hessian experiments. For example, [1] (refer Appendix D) uses 10% of training data for Hessian Eigenvalue calculation. We have added these details in Appendix Section D (Hessian Analysis) of the updated version of the draft.
> >
> > [9] Yao, Z., Gholami, A., Keutzer, K., & Mahoney, M. W. (2020, December). Pyhessian: Neural networks through the lens of the hessian. In 2020 IEEE International Conference on Big Data (Big Data) (pp. 581-590). IEEE.
> >
> > > SDAT requires an extra gradient step in order to estimate the smoothing penalty in Eq. 7. How does this affect the computational cost of SDAT and how SDAT stands in comparison with the considered baselines?
> >
> > Yes, SDAT requires an extra gradient step. However, there have been recent works [8] that make SAM much more efficient. Also, the other baselines have their overhead with additional loss computation. (For example, SRDC). Furthermore, our method is conceptually simple and can be adapted to different tasks. We report the wall clock times of the CDAN w/ SDAT and MDD on a split of Office-Home dataset. All the experiments were run on the same GPU (RTX 2080 Ti). It can be seen that the computational cost of CDAN w/ SDAT and MDD are comparable. Efficient-SAM [8] mentions that it requires only a 40% overhead over the baseline and incorporating it would further reduce the wall clock time of CDAN w/ SDAT (~ 3hours 20 minutes).
> >
> > | Method      | Wall Clock Time |
> > | :---        | ---: |
> > | CDAN     | 2 hours 30 minutes  |
> > | CDAN w/ SDAT | 4 hours 33 minutes |
> > | MDD     |  4 hours 40 minutes |
> >
> > [8] Du, J., Yan, H., Feng, J., Zhou, J. T., Zhen, L., Goh, R. S. M., & Tan, V. Y. (2021). Efficient Sharpness-aware Minimization for Improved Training of Neural Networks. arXiv preprint arXiv:2110.03141.
> >
> > > Moreover, the relevance of the presented generalization bound is also unclear to me at this point.`
> >
> > By the generalization bound (Theorem 3) we hope to establish consistency of the procedure which establishes that our procedure is sound and consistent (and similar to previous work in literature).

---

> > > ### Author Response · Authors · 2021-11-15
> > > **Response to Reviewer rxW2 [3/3]**
> > >
> > > > Abstract: “combination of classification and adversarial terms”. In the considered setting, the adversarial term of the loss also stems from a classification task (domain discrimination), therefore, it is confusing to the denominate both component losses in such a way. I suggest referring to the loss term related to classifying labels regarding a specific task as task loss.
> > >
> > > We thank the reviewer for this suggestion. We agree that "task loss" would be better. We have changed it in the paper.
> > >
> > > > Table 1 caption: what exactly “sophisticated” means here?
> > >
> > > We refer to the other DA algorithms as sophisticated because of the additional loss terms introduced in them. Most of the times, these terms cannot be easily extended to other tasks like dense estimation and semantic segmentation. Whereas, the proposed SDAT can be extended to object detection as shown in the paper.
> > >
> > > > Foundational literature on domain adaptation is missing in the related work section, for example, Ben-David et al. (2007) and Long et al. (2016).
> > >
> > > We apologise for missing out on these references. We will add these references in the updated draft.
> > >
> > > > Theorems 1/3: please precisely define what is the ideal classifier h*.
> > >
> > > The ideal classifier is defined as the classifier with the joint lowest source risk and target risk.
> > >
> > > We have incorporated all suggestions including minor suggestions in the updated version, which have been useful in increasing the readability of our draft. Please let us know if you require any additional clarifications.

---

> > > > ### Comment · Reviewer_rxW2 · 2021-11-19
> > > > **Further comments**
> > > >
> > > > Dear authors, thank you for your effort in writing a detailed response to my comments and for addressing many of my concerns.
> > > >
> > > > Although I appreciate the new results presented in Table S9, my concerns with the empirical validation of SDAT still remain. I understand that the authors attempted to run experiments with the same environment for all the compared approaches, but this fact alone does not make the conclusions drawn from the results stronger as it is not possible to tell whether the same slight changes in the experimental conditions (e.g. random seed, dataset split) would lead to drastically different findings. Moreover, I am aware that domain adaptation/generalization experimental settings are expensive due to the multiple combinations of domains that can be tested. However, I believe that from a scientific method standpoint, it is better to employ the available computational budget to provide robust and significant conclusions to experiments in fewer test cases, rather than presenting a potentially inconclusive evaluation of the proposed approach in several experimental conditions. Given that, it is necessary to at least acknowledge and discuss the limitations of the empirical findings of the paper and make it clear that the presented results might not lead to strong conclusions due to a lack of statistical significance. On a related note, I couldn’t find in the main paper any discussion regarding the results presented in Table S9 (in fact, this table was not even mentioned in the main paper).
> > > >
> > > > Regarding the generalization bound presented in Theorem 3, I appreciate the clarification that this result was presented solely as a means of showing the soundness of the proposed approach. However, this is not clear in the first version of the paper and the current manuscript doesn’t contain any mention to this point as well.
> > > >
> > > > Even though I still have important concerns regarding this submission, I believe many of the points I raised in my review were addressed by the authors' comments. All in all, I decided to raise my score to 5.

---

> > > > > ### Author Response · Authors · 2021-11-23
> > > > > **Thank you for the feedback! Response to Feedback [1/2]**
> > > > >
> > > > > Thanks for your invaluable comments and suggestions. We sincerely thank you for taking the time to take a closer look at the rebuttal and increasing the score. We have incorporated the suggestions in the revised manuscript and highlighted them in **purple**.
> > > > >
> > > > > > Random Seeds
> > > > >
> > > > >
> > > > > Yes, we do acknowledge that random seeds can change the results. However, we would like to clarify that for the largest available benchmark of  DomainNet (0.6 million images ~ ½ ImageNet) the variance in increase is smaller in comparison to Office datasets. For further establishing the soundness, we provide results on 3 different random seeds for 6 different splits (across 5 different domains) in the table below (Table S9 in paper). This leads to a total of 36 experiments (~20 hrs each) (i.e., 18 for CDAN baseline and 18 for proposed CDAN w/ SDAT), where we consistently observe that our method significantly improves over baseline. In each case, we observe that the average increase across 3 random seeds is similar to the increase we report in Table 2 (single fixed seed), which increases the reliability of reported results. Furthermore, in multiple recent published works too, performance of only a single run (similar to us) is reported as a reliable estimate due to large size of DomainNet dataset[1,2].
> > > > >
> > > > >
> > > > > | Split      | CDAN| CDAN w/ SDAT | Reported Increase | Average Increase |
> > > > > | :---       |   :----:   |      :----:   |   :----:   |     ---: |
> > > > > | clp -> pnt     | 38.9  ± 0.1|  41.5 ± 0.3  | +2.6 | +2.6  |
> > > > > | skt -> rel     | 55.1 ± 0.2      |  57.1 ±0.1 | +2.2 | +2.0   |
> > > > > | pnt -> clp     |  44.5 ± 0.3      |  47.1 ± 0.3  | +3.4 | +2.6   |
> > > > > | rel -> skt     |  42.4 ± 0.4      |  43.9 ± 0.1  | +1.6 | +1.5   |
> > > > > | clp -> skt     |  44.9 ± 0.2      |  47.3 ± 0.1  | +2.3 | +2.4   |
> > > > > | inf -> clp     |  31.4 ± 0.5      |  34.2 ± 0.3  | +2.3 | +2.7   |
> > > > >
> > > > >
> > > > > We also run experiments on Office-Home, where due to only 44 img/cls we observe some variance in the accuracy of the CDAN baseline. However, still in each case, we were able to observe improvement with SDAT over baseline consistently. We have added our experimental results in Table S11 (in App. J).
> > > > >
> > > > > [1] Jin, Y., Wang, X., Long, M., & Wang, J. (2020, August). Minimum class confusion for versatile domain adaptation. In European Conference on Computer Vision (pp. 464-480). Springer, Cham.
> > > > >
> > > > > [2] Prabhu, V., Khare, S., Kartik, D., & Hoffman, J. (2021). Sentry: Selective entropy optimization via committee consistency for unsupervised domain adaptation. In Proceedings of the IEEE/CVF International Conference on Computer Vision (pp. 8558-8567).
> > > > >
> > > > > > Dataset Splits
> > > > >
> > > > >
> > > > > In the case of DomainNet, we use the predefined validation, training, and test set provided by the dataset creators. The same is the case with other datasets. We do not create any random dataset split by ourselves; hence, results are always on a fixed split of data for all methods.
> > > > >
> > > > > Also, for results on adaptation from different domains, we observe that the avg. results  for 3 trials do not differ much from the single seed in Table above. This provides evidence that avg. results across three on other splits may be similar with high probability, which we are unable to run multiple times due to rebuttal time constraints.
> > > > >
> > > > > In a concurrent work [3] due to large computational expense, the authors report the median of the accuracy of the last few checkpoints for DomainNet as reliable performance estimate. Following them, we also use a technique of reporting the median validation accuracy of the last five checkpoints in Table S10. We find that across *all dataset splits* the median accuracy of the proposed SDAT significantly improves over baseline. This establishes that our method’s best result is not an outlier and produces better results across all the later epochs.
> > > > >
> > > > > [3] Berthelot, D., Roelofs, R., Sohn, K., Carlini, N., & Kurakin, A. (2021). AdaMatch: A Unified Approach to Semi-Supervised Learning and Domain Adaptation. arXiv preprint arXiv:2106.04732.
> > > > >
> > > > > > Stability of Results:
> > > > >
> > > > > For further analysis, we also provide plots of validation accuracy across three different seeds for 6 splits in Fig. S1. We observe that our method achieves better results in all cases without overlapping confidence intervals in later epochs.
> > > > >
> > > > > > Acknowledgment of Statistical Significance
> > > > >
> > > > > We apologize for	missing this out, we have now mentioned that only on a subset of results the experiment is repeated 3 times (with different random seeds) due to computational constraints. However, we observe across 6 splits (run 3 times) that due to the large size of DomainNet there is  less variance in reported vs avg. results (in Table above), hence the single run numbers are a reliable estimate of performance improvement. We have also referred to the section of statistical significance testing in the main paper, thanks for the suggestion. We will add more results and explanations in the main draft in the camera ready version.

---

> > > > > > ### Author Response · Authors · 2021-11-23
> > > > > > **Thank you for the feedback! Response to Feedback [2/2]**
> > > > > >
> > > > > >
> > > > > > > Missing Explanation for Theorem 3
> > > > > >
> > > > > > We apologize for missing out on improving the description of Theorem 3, we have now stated the point of consistency clearly in the main paper.
> > > > > >
> > > > > > We sincerely thank you for your comments and suggestions, which have immensely improved our draft. We would be really grateful and thankful if you can go through our response and reconsider our contribution.

---

> > > > > > > ### Comment · Reviewer_rxW2 · 2021-11-29
> > > > > > > **Further comments**
> > > > > > >
> > > > > > > Dear authors, thank you for your effort in the rebuttal and for the extra experiments. In light of this new empirical evidence to support some of the claims in the manuscript, I increased the scores of "Correctness" and "Empirical Novelty And Significance" from 2 to 3.

---

### Author Response · Authors · 2021-11-19
**Summary of Changes in Draft**

We thank all the reviewers for their time and valuable feedback on our paper. We are happy that reviewers find our work interesting, relevant, novel, empirically well supported, and useful for both detection and classification tasks. We have updated the paper with the following key changes after incorporating suggestions from all the reviewers.

List of changes (All the changes are **highlighted in blue**):
1. "classification loss" has been changed to "task loss" as suggested by Reviewer rxW2.
2. Added a section on "Optimum $\rho$ value" (Appendix I) and "Reliability of Empirical Results" (Appendix J).
3. Improved the motivation section as per suggestions (Section 2).
4. Add more details in the Hessian Analysis section (Appendix D)
5. Added a table to clearly demonstrate the impact of smoothing each component (Figure 4 C)
6. Improved Grammar and Writing.


We have completed posting our initial responses and eagerly look forward to their further comments and feedback.

---

### Decision · Program_Chairs · 2022-01-20

**Decision:**

Reject

**Comment:**

This paper studies the loss landscape of domain adversarial neural networks for domain adaptation. First, the authors show that smooth minima with respect to adversarial loss leads to sub-optimal generalization on the target domain. Then, they suggest to enforce smoothness only with respect to the task loss. 3 reviewers are on negative sides, and 1 reviewer is on a positive side. All negative reviewers interacted with authors in the discussion period. After the discussion period, even the positive reviewer agreed negative comments of other reviewers and declined to champion this paper.

AC thinks that this is a borderline paper; the proposed claims (both theoretic and empirical) are interesting and there is no critical weakness of this paper. However, the results are not super excited, as evidenced by 3 negative reviewers. In particular, AC agrees with negative comments of reviewers on limited novelty and lack of strong theoretical motivation for the proposed scheme. AC also thinks the performance improvements in experiments are not that significant. Furthermore, the problem scope is narrow, i.e., the authors study a certain property (smoothness) of a special algorithm (adversarial training) for domain adaptation that is a particular way for domain generalization. Hence, the impact to the community can be not significant as well. Considering all aspects, AC is a bit toward to suggesting rejection.